# Diffusion with Synthetic Features: Feature Imputation for Graphs with Partially Observed Features

## Abstract

In this paper, we tackle learning tasks on graphs with missing features, improving the applicability of graph neural networks to real-world graph-structured data. Previous diffusion-based imputation methods overlook the presence of channels with low-variance features, and these channels contribute very little to the performance in graph learning tasks. To overcome this issue, we propose a new diffusion-based imputation scheme using synthetic features in addition to observed features. The proposed scheme first identifies channels with low-variance features via pre-diffusion and generates a synthetic feature for a randomly chosen node in each low-variance channel. Then, our diffusion process spreads the synthetic features widely while considering observed features simultaneously. Extensive experiments on graphs with various rates of missing features demonstrate the effectiveness of our scheme, achieving state-of-the-art performance in both semi-supervised node classification and link prediction.

## 1 Introduction

Graph neural networks (GNNs) have achieved significant successes in graph learning tasks such as node classification (Kipf & Welling, 2016a; Veličković et al., 2017) and link prediction (Kipf & Welling, 2016b; Salha et al., 2019). In the real world, since a wide range of data contains entities with relations, these data can be represented in graphs and many problems are formulated as graph learning tasks (Wu et al., 2022; Liao et al., 2021). However, real-world graph-structured data often include missing features for various reasons (*e.g.*, private information in social networks and measurement failure), which hinders GNNs from being directly applied to the real-world data. Therefore, applying GNNs to graphs with missing features has received great attention as a task termed graph learning task with missing features (Chen et al., 2020; Taguchi et al., 2021).

Recent diffusion-based imputation approach (Rossi et al., 2022; Um et al., 2022) imputes missing features by diffusing observed features along edges channel-wisely. The diffusion-based methods demonstrate the following two advantages against conventional neural-network-based imputation methods (Monti et al., 2017; Chen et al., 2020): 1) superior performance and 2) fast imputation without learnable parameters. A crucial observation is that the accurate reconstruction of missing features does not necessarily result in good performance in graph learning tasks (Um et al., 2022). That is, producing features that are close to their original values and generating features that lead to good performance in graph learning tasks are distinct tasks.

However, the diffusion-based methods overlook the presence of channels with low-variance features as shown in Figure 1. When all observed features within a low-variance channel have almost the same values, the diffusion process fills all missing features in the channel with nearly the same values. In our work, we empirically discover these channels referred to as *low-variance channels* and theoretically prove that a zero-variance channel is made when values of all observed features in the channel are the same. The channels with nearly the same feature values across entire nodes contribute very little to performance in graph learning tasks which require distinct representations of nodes or node pairs.

To increase feature variances of low-variance channels, we propose a novel diffusion-based imputation scheme called Feature Imputation with Synthetic Features (FISF). FISF generates synthetic features different from observed features and injects the synthetic features into a randomly chosen

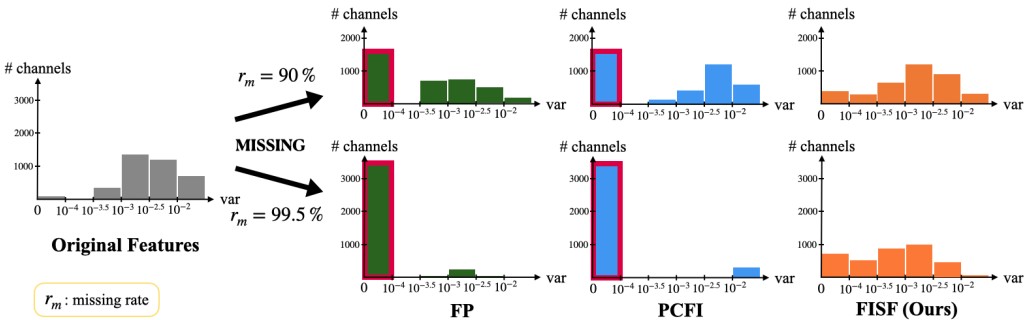

Figure 1: Distributions of variances for each feature channel. The distributions for imputation methods are calculated from imputed matrices for the CiteSeer dataset with $90\%/99.5\%$ missing features. While existing diffusion-based imputation methods (FP and PCFI) produce output with many low-variance channels (outlined in red), FISF can address the problem of low-variance channels.

node in each low-variance channel. Then, FISF spreads the generated synthetic features and makes the channels deviate from low variances. We show that GNNs with an imputed feature matrix from FISF demonstrate remarkable performance in graph learning tasks with missing features.

Our key contributions are summarized as follows: 1) We discover low-variance channels in outputs from diffusion-based imputation methods, which provide little assistance in performance on graph learning tasks. 2) This work is the first attempt for feature imputation using synthetic features for graph learning tasks with missing features. 3) We demonstrate that the use of synthetic features significantly enhances performance on graph learning tasks under various missing feature settings.

## 2 RELATED WORK

### 2.1 LEARNING ON GRAPHS WITH MISSING FEATURES

Dealing with missing data has long been an active research field in machine learning (Allison, 2009; Troyanskaya et al., 2001). Methods for handling missing data in graph-structured data can be categorized into three groups.

(i) *GNN Architecture.* Several methods propose new GNN architectures to perform learning tasks on graphs with missing features. GCN for missing features (GCNMF) (Taguchi et al., 2021) combines a GCN (Kipf & Welling, 2016a) layer with a Gaussian mixture model that represents missing features. Jiang & Zhang (2020) develops a message passing layer that only aggregates known features. Graph feature neural network (GRAFENNE) (Gupta et al., 2023) consists of three-phase message-passing layers to address heterogeneous and dynamic features. However, these methods, with their specially designed layers, cannot take advantage of the off-the-shelf GNN models.

(ii) *Reconstruction.* Reconstruction-based methods train models by minimizing reconstruction error between the observed features and their reconstructed values. Recurrent Multi-Graph CNN (RMGCNN) leverages recurrent neural networks to complete a feature matrix (Monti et al., 2017). Structure-attribute-transformer (SAT) (Chen et al., 2020) models the joint distribution of graph structures and node features. Max-entropy graph autoencoder (MEGAE) (Gao et al., 2023) maximizes the entropy of latent features in autoencoders to alleviate the spectral concentration problem. While these methods aim to accurately reconstruct missing features, achieving accurate reconstructed features does not necessarily guarantee high performance in downstream tasks (Um et al., 2022).

(iii) *Diffusion.* Diffusion-based methods impute missing features by diffusing known features along edges. Feature propagation (FP) (Rossi et al., 2022) iteratively propagates known features channel-wisely and fills in missing features. Pseudo-confidence-based feature imputation (PCFI) (Um et al., 2022) calculates pseudo-confidence of each feature value and leverages pseudo-confidence as the importance of feature values during diffusion. These methods tend to make missing features very similar to each other when a few observed features are highly similar, resulting in minimal feature differences between nodes. Our approach encourages distinct features between nodes, which can further enhance the performance of downstream GNNs in graph learning tasks.

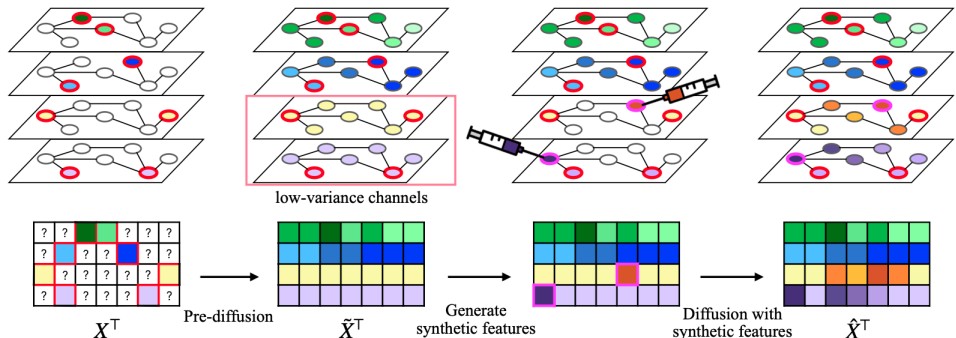

Figure 2: A brief overview of feature imputation with synthetic features (FISF). First, pre-diffusion constructs a full feature matrix $\tilde{\boldsymbol{X}}$ by imputing missing features via channel-wise diffusion. Then, we inject synthetic features into a missing node in each low-variance channel of $\tilde{\boldsymbol{X}}$. Finally, diffusion with synthetic features produces $\hat{\boldsymbol{X}}$ which is a final output of FISF. $\hat{\boldsymbol{X}}$ is fed to a downstream GNN which performs a given graph learning task.

## 2.2 OTHER RELATED WORK

To spread synthetic features widely, we assign different importance based on distance encoding. Distance encoding is a technique that utilizes graph-distance measures (*e.g.*, shortest path distance, generalized PageRank scores (Li et al., 2019)) measured between a node and a designated node set. You et al. (2019) proposes an aggregation scheme using the computed distance of a given node from sampled anchor node sets. Zhang & Chen (2018) and Li et al. (2020) leverage encoded distance as extra node features for link prediction. Position-aware graph neural network (P-GNN) (Zhang et al., 2021) unifies several techniques including distance encoding into a labeling trick. Poisson learning (Calder et al., 2020) addressing a problem in Laplacian learning is also relevant to our work. However, the problems being addressed are different, and the causes of each problem are also different. In Calder et al. (2020), proposed Poisson learning is a feature-agnostic method that only propagates given labels like LP (Zhuŕ & GhahramaniŕH, 2002), tackling semi-supervised classification. Furthermore, the problem addressed in Calder et al. (2020) arises from the very narrow area of a localized spike, generated by the propagation of a given label. The problem assumes having a wide variety of labels evenly distributed despite very low label rates. However, we discover and address the problem of low-variance channels caused by nearly identical observed values with a feature channel. We provide theoretical proof about the cause of the problem of low-variance channels.

## 3 NOTATION AND PROBLEM DEFINITION

An undirected connected graph can be represented as $\mathcal{G} = (\mathcal{V}, \mathcal{E}, \boldsymbol{A})$ where $\mathcal{V} = \{v_1, \ldots, v_N\}$ is the set of $N$ nodes, $\mathcal{E}$ is the edge set, and $\boldsymbol{A} \in \{0,1\}^{N \times N}$ is an adjacency matrix. $\boldsymbol{X} = [x_{i,a}] \in \mathbb{R}^{N \times F}$ denotes a node feature matrix where $F$ is the number of feature channels and $x_{i,a}$ represents the $a$-th channel feature value of $v_i$.

Let $d(v_i, v_j|\boldsymbol{A})$ be the shortest path distance between the $i$-th node and the $j$-th node on $\mathcal{G}$ with $\boldsymbol{A}$. Then, we define a function $d_{set}(\cdot)$ as $d_{set}(v_i|\mathcal{V}', \boldsymbol{A}) = \min_{v_j \in \mathcal{V}'} d(v_i, v_j|\boldsymbol{A})$ where $\mathcal{V}' \subseteq \mathcal{V}$. That is, we use $d_{set}(v_i|\mathcal{V}', \boldsymbol{A})$ to denote the shortest path distance between the $i$-th node and its nearest node in a node set $\mathcal{V}' \subseteq \mathcal{V}$ on $\mathcal{G}$ with $\boldsymbol{A}$.

Partially observed/known features mean that $\boldsymbol{X}$ has missing elements. $\mathcal{V}_k^{(a)}$ denotes a set of nodes whose $a$-th channel feature values are known. $\mathcal{V}_u^{(a)}$ denotes a set of nodes whose $a$-th channel feature values are missing/unknown (*i.e.*, $\mathcal{V}_u^{(a)} = \mathcal{V} \setminus \mathcal{V}_k^{(a)}$). We refer to $\mathcal{V}_k^{(a)}$ and $\mathcal{V}_u^{(a)}$ as source nodes and missing nodes, respectively. By rearranging the whole nodes based on whether the feature value is known or not for each channel, the whole features and the adjacency matrix for the $a$-th channel can be written as

$$\boldsymbol{x}^{(a)} = \begin{bmatrix} \boldsymbol{x}_k^{(a)} \\ \boldsymbol{x}_u^{(a)} \end{bmatrix}, \qquad \boldsymbol{A}^{(a)} = \begin{bmatrix} \boldsymbol{A}_{kk}^{(a)} & \boldsymbol{A}_{ku}^{(a)} \\ \boldsymbol{A}_{uk}^{(a)} & \boldsymbol{A}_{uu}^{(a)} \end{bmatrix}, \tag{1}$$

where $\boldsymbol{x}^{(a)}$, $\boldsymbol{x}_k^{(a)}$, and $\boldsymbol{x}_u^{(a)}$ are column vectors for the $a$-th channel. $\boldsymbol{A}^{(a)}$ and $\boldsymbol{A}$ represent the same graph structure although the node order of $\boldsymbol{A}^{(a)}$ is rearranged from $\boldsymbol{A}$. We use $\boldsymbol{B}_{:,z}$ to denote the $z$-th column of a matrix $\boldsymbol{B}$.

We tackle a problem of graph learning tasks containing missing features, where our goal is to achieve maximum performance in downstream learning tasks. Formally, graph learning tasks containing missing features can be expressed as

$$\hat{\boldsymbol{Y}} = f(\{\boldsymbol{x}_k^{(a)}\}_{a=1}^F, \boldsymbol{A}) \tag{2}$$

where $\hat{\boldsymbol{Y}}$ denotes a prediction for desired output of a given task. Here, $f$ is a function to find in the problem. Like other imputation methods tackling the problem, we decompose $f$ into two steps as $f = g_\theta \circ h$. Here, $h$ is a feature imputation scheme and $g_\theta$ is an off-the-shelf GNN model using a full feature matrix obtained via $h$.

## 4 PROPOSED METHOD

### 4.1 OVERVIEW OF FISF

We present an imputation scheme called feature imputation with synthetic features (FISF), which minimizes performance degradation in graph learning tasks despite high rates of missing features. Figure 2 shows a brief overview of FISF which consists of two diffusion stages: *pre-diffusion* and *diffusion with synthetic features*. Using a pre-imputed feature matrix obtained via pre-diffusion (see Section 4.2), we calculate the variance of features for each channel. We then create a synthetic feature in each low-variance channel (see Section 4.3). The second diffusion stage updates the features in low-variance channels by spreading the synthetic features widely (see Section 4.4). The stage produces a final output feature matrix of FISF, which is fed to $g_\theta$ to perform downstream tasks.

### 4.2 PRE-DIFFUSION

We adopt channel-wise inter-node diffusion in PCFI (Um et al., 2022) as pre-diffusion. For notational convenience, we temporarily rearrange whole nodes channel-wisely as described in Section 3. Specifically, for the $a$-th channel, we reorder the nodes in the order of $\mathcal{V}_k^{(a)}$ and $\mathcal{V}_u^{(a)}$, *i.e.*, $\boldsymbol{x}^{(a)}$ and $\boldsymbol{A}^{(a)}$ are made by reordering $\boldsymbol{A}$. After the diffusion is completed, we restore the node ordering to the original one.

The channel-wise inter-node diffusion calculates and utilizes pseudo-confidence (PC) (Um et al., 2022), which acts as the importance of each feature value during the diffusion. We use $\boldsymbol{S}_{i,a}$ to denote the shortest path distance between the $i$-th node and its nearest source node for the $a$-th channel, *i.e.*, $\boldsymbol{S}_{i,a} = d_{set}(v_i | \mathcal{V}_k^{(a)}, \boldsymbol{A}^{(a)})$. We let $\tilde{X}$ be a pre-imputed feature matrix via pre-diffusion. Then, following Um et al. (2022), PC ($\xi_{i,a}$) of $\tilde{x}_{i,a}$ is assigned by $\xi_{i,a} = \alpha^{\boldsymbol{S}_{i,a}}(0 < \alpha < 1)$ where $\alpha$ is a hyper-parameter. Thereafter, the transition matrix for the pre-diffusion is built by a weighted adjacency matrix $\boldsymbol{W}^{(a)} \in \mathbb{R}^{N \times N}$ given by

$$\boldsymbol{W}_{i,j}^{(a)} = \begin{cases} \xi_{j,a}/\xi_{i,a} & \text{if } \boldsymbol{A}_{i,j}^{(a)} = 1 \\ 0 & \text{if } \boldsymbol{A}_{i,j}^{(a)} = 0, \end{cases} \tag{3}$$

where $\boldsymbol{W}_{i,j}^{(a)}$ takes a role of message passing strength from the $j$-th node to the $i$-th node in the pre-diffusion. For a row-stochastic transition matrix, we normalize $\boldsymbol{W}^{(a)}$ to $\overline{\boldsymbol{W}}^{(a)} = (\boldsymbol{D}^{(a)})^{-1}\boldsymbol{W}^{(a)}$ where $\boldsymbol{D}^{(a)}$ is a diagonal matrix with diagonal entries $\boldsymbol{D}_{ii}^{(a)} = \sum_j \boldsymbol{W}_{i,j}$. Then, to preserve the known features $\boldsymbol{x}_k^{(a)}$ during the pre-diffusion, we replace the first $|\mathcal{V}_k^{(a)}|$ rows in $\overline{\boldsymbol{W}}$ with one-hot vectors indicating $\mathcal{V}_k^{(a)}$. As a result of the replacement, we attain the pre-diffusion transition matrix $\widetilde{\boldsymbol{W}}^{(a)}$ expressed by

$$\widetilde{\boldsymbol{W}}^{(a)} = \begin{bmatrix} \boldsymbol{I}_{kk} & \boldsymbol{0}_{ku} \\ \boldsymbol{W}_{uk}^{(a)} & \overline{\boldsymbol{W}}_{uu}^{(a)} \end{bmatrix}, \tag{4}$$

where $\boldsymbol{I}_{kk} \in \mathbb{R}^{|\mathcal{V}_k^{(a)}| \times |\mathcal{V}_k^{(a)}|}$ is an identity matrix and $\boldsymbol{0}_{ku} \in \mathbb{R}^{|\mathcal{V}_k^{(a)}| \times |\mathcal{V}_u^{(a)}|}$ is a zero matrix.

The pre-diffusion is implemented by iterative propagation steps using $\widetilde{\boldsymbol{W}}^{(a)}$ as

$$
\begin{aligned}
\tilde{\boldsymbol{x}}^{(a)}(t) &= \widetilde{\boldsymbol{W}}^{(a)} \tilde{\boldsymbol{x}}^{(a)}(t-1), \ \ t = 1, \cdots, K; \\
\tilde{\boldsymbol{x}}^{(a)}(0) &= \begin{bmatrix} \boldsymbol{x}_k^{(a)} \\ \boldsymbol{0}_u \end{bmatrix},
\end{aligned}
\tag{5}
$$

where $\tilde{\boldsymbol{x}}^{(a)}(t)$ is an imputed feature vector after $t$ propagation steps and $\boldsymbol{0}_u$ is a zero vector with a length of $|\mathcal{V}_u^{(a)}|$. After $K$ propagation steps, we obtain $\tilde{\boldsymbol{x}}^{(a)}(K)$. As $K \to \infty$, the recursion converges and $\tilde{\boldsymbol{x}}^{(a)}(K)$ reaches a steady state (see the proof in Appendix A)). Based on the proof that initial values for $\boldsymbol{x}_u^{(a)}$ do not affect the steady state, we initialize $\boldsymbol{x}_u^{(a)}$ with zeros (*i.e.*, $\boldsymbol{0}_u$). We use $\tilde{\boldsymbol{x}}^{(a)}(K)$ with large enough $K$ to approximate the steady state.

We rearrange $\{\tilde{\boldsymbol{x}}^{(a)}(K)\}_{a=1}^F$ in the original order to reorder the nodes considering the synthetic features in the second diffusion stage. Then, by stacking the originally ordered vectors in $\{\tilde{\boldsymbol{x}}^{(a)}(K)\}_{a=1}^F$ along the channels, we obtain a pre-imputed feature matrix $\tilde{\boldsymbol{X}}$ which is an output of the pre-diffusion.

### 4.3 SYNTHETIC FEATURE GENERATION

When all given known features in the $a$-th channel (*i.e.*, elements in $\boldsymbol{x}_k^{(a)}$) have the same value $c$, $\lim_{t \to \infty} \tilde{\boldsymbol{x}}^{(a)}(t)$ becomes a vector where entire elements are $c$ (see the proof in Appendix B)). We refer to a channel with the same or nearly the same feature values as a *low-variance channel*. The low-variance channel does not contribute to distinguishing nodes. In semi-supervised node classification, distinctive node representations are crucial to classify nodes into multiple classes. In the case of link prediction, the same representation across nodes also makes the representations of node pairs the same. Therefore, we aim to make imputed features in that channel become distinctive across nodes by injecting a synthetic feature that acts as a known feature.

We first identify low-variance channels to inject synthetic features. We calculate the variance of $\tilde{\boldsymbol{X}}_{:,a}$ (*i.e.*, pre-imputed feature values in the $a$-th channel) for all $a \in \{1, \ldots, F\}$. Then $r\%$ of channels are selected in order of lowest to highest variance, where $r$ is a hyper-parameter between 0 and 100. $\mathbb{F}_l$ denotes the set of low-variance channel indices. For each channel in $\mathbb{F}_l$, we randomly choose one node with a missing feature to inject a synthetic feature. For a selected node $v_s^{(b)}$ in a channel $b \in \mathbb{F}_l$, we inject a synthetic feature with randomly sampled value $\mathrm{x}_s^{(b)}$ from a uniform distribution on $[0, 1]$. Consequently, $|\mathbb{F}_l|$ number of synthetic feature values are injected and $\{(v_s^{(b)}, \mathrm{x}_s^{(b)})\}_{b \in \mathbb{F}_l}$ is combined with the result of pre-diffusion ($\tilde{\boldsymbol{X}}$) for the second diffusion stage called diffusion with synthetic features.

### 4.4 DIFFUSION WITH SYNTHETIC FEATURES

Diffusion with synthetic features (DSF) produces $\hat{\boldsymbol{X}} = [\hat{x}_{i,a}] \in \mathbb{R}^{N \times F}$ which is a final output of FISF. DSF receives $\tilde{\boldsymbol{X}}$ from the pre-diffusion and $\{(v_s^{(b)}, \mathrm{x}_s^{(b)})\}_{b \in \mathbb{F}_l}$. Then DSF updates $\tilde{\boldsymbol{X}}$ by replacing features in the low-variance channels (*i.e.*, $\tilde{\boldsymbol{X}}_{:,b}$ for all $b \in \mathbb{F}_l$). The purpose of DSF is to increase the variance of low-variance channels by using synthetic features.

DSF treats a synthetic feature $\mathrm{x}_s^{(b)}$ as known features $\boldsymbol{x}_k^{(b)}$ during diffusion. Then the updated known node set becomes $\mathcal{V}_{k^*}^{(b)} = \mathcal{V}_k^{(b)} \cup \{v_s^{(b)}\}$. Thus the updated unknown node set becomes $\mathcal{V}_{u^*}^{(b)} = \mathcal{V}_u^{(b)} \setminus \{v_s^{(b)}\}$. That is, $v_s^{(b)}$ is moved from $\mathcal{V}_u^{(b)}$ to $\mathcal{V}_{k^*}^{(b)}$. Similar to pre-diffusion, we first temporarily reorder all the nodes in the order of $\mathcal{V}_{k^*}^{(b)}$ and $\mathcal{V}_{u^*}^{(b)}$. By reordering, features and the adjacency matrix in the $b$-th channel in $\mathbb{F}_l$ can be expressed as

$$
\boldsymbol{x}^{(b)} = \begin{bmatrix} \boldsymbol{x}_{k^*}^{(b)} \\ \boldsymbol{x}_{u^*}^{(b)} \end{bmatrix}, \qquad \boldsymbol{A}^{(b)} = \begin{bmatrix} \boldsymbol{A}_{k^*k^*}^{(b)} & \boldsymbol{A}_{k^*u^*}^{(b)} \\ \boldsymbol{A}_{u^*k^*}^{(b)} & \boldsymbol{A}_{u^*u^*}^{(b)} \end{bmatrix},
\tag{6}
$$

where $\boldsymbol{x}_{k^*}^{(b)}$ and $\boldsymbol{x}_{u^*}^{(b)}$ are column vectors and $\boldsymbol{x}_{k^*}^{(b)}$ contains $\mathrm{x}_s^{(b)}$. The length of $\boldsymbol{x}_{k^*}^{(b)}$ and $\boldsymbol{x}_{u^*}^{(b)}$ are $|\mathcal{V}_k^{(b)}| + 1$ and $|\mathcal{V}_u^{(b)}| - 1$, respectively.

The preparations above are the same as the pre-diffusion, except for assuming $\mathrm{x}_s^{(b)}$ as a known feature. However, simply diffusing features of $\mathcal{V}_{k^*}^{(b)}$ as pre-diffusion results in $\mathrm{x}_s^{(b)}$ influencing only its surroundings. This is because not only $\mathrm{x}_s^{(b)}$ but also known features with nearly the same values diffuse. For example, if a given graph has $10,000$ nodes and $90\%$ features are missing in the $b$-th channel, there exist $1,000$ known features with nearly the same feature values in the channel. Known features spread to their surrounding features through diffusion and make the surrounding features be similar to their own value. Thus, it is hard for $\mathrm{x}_s^{(b)}$ to have a wide influence across nodes. This issue hinders the channel from deviating from a low variance since most of the features become nearly the same value.

To overcome the issue, we design DSF to give more influence to synthetic features than that of known features. For the wide diffusion of $\mathrm{x}_s^{(b)}$, we leverage the shortest path distance from $v_s^{(b)}$. We measure the shortest path distance from $v_s^{(b)}$ to all nodes in $\mathcal{V}$. Formally, we use $\boldsymbol{S}_{i,b}^s$ to denote $d(v_i, v_s^{(b)} | \boldsymbol{A}^{(b)})$ and measure $\boldsymbol{S}_{i,b}^s$ for all $v_i \in \mathcal{V}$.

Then the PC $\xi_{i,a}^s$ of $\hat{x}_{i,a}$ is computed based on the shortest path distance from only the synthetic node $v_s^{(b)}$, not from the whole known nodes. That is, $\xi_{i,a}^s$ is defined by $\xi_{i,a}^s = \beta^{\boldsymbol{S}_{i,a}^s}(0 < \beta < 1)$ where $\beta$ is a hyper-parameter. As $v_i$ is positioned closer to $v_s^{(b)}$, $\xi_{i,a}^s$ increases. We also use usual PC ($\xi_{i,b}^*$) based on distances from the whole known nodes $\mathcal{V}_{k^*}^{(b)}$ containing $v_s^{(b)}$. We calculate $\boldsymbol{S}_{i,b}^* = d_{set}(v_i | \mathcal{V}_{k^*}^{(b)}, \boldsymbol{A}^{(b)})$ and obtain PC calculated by $\xi_{i,b}^* = \alpha^{\boldsymbol{S}_{i,b}^*}(0 < \alpha < 1)$. While both $\xi_{i,b}$ and $\xi_{i,b}^*$ play a role as the importance of each feature value, $\xi_{i,b}$ is determined by the distance from only synthetic node $v_s^{(b)}$ in contrast to $\xi_{i,b}^*$ considering the distances from whole known nodes $\mathcal{V}_{k^*}^{(b)}$. Using the PCs, we define a weighted adjacency matrix $\boldsymbol{M}^{(b)} \in \mathbb{R}^{N \times N}$ by

$$\boldsymbol{M}_{i,j}^{(b)} = \begin{cases} \dfrac{\xi_{j,b}^*}{\xi_{i,b}^*} \cdot \dfrac{\xi_{j,b}^s}{\xi_{i,b}^s} & \text{if } \boldsymbol{A}_{i,j}^{(b)} = 1 \\ 0 & \text{if } \boldsymbol{A}_{i,j}^{(b)} = 0. \end{cases} \tag{7}$$

$\boldsymbol{M}_{i,j}^{(b)}$ is the strength of a message passing from the $j$-th node to the $i$-th node in the DSF.

The term $\xi_{j,b}^*/\xi_{i,b}^*$, strengthens a message passing from a high-PC feature to a low-PC feature as in the pre-diffusion (see Eq. 3). However, different from the pre-diffusion, the synthetic feature of $v_s^{(b)}$ is considered as one of the nodes in $\mathcal{V}_k^{(b)}$. Thus the influence of the synthetic feature is very weak compared to the many observed similar features. To widely spread the synthetic feature, we introduce the term $\xi_{j,b}^s/\xi_{i,b}^s$, which strengthens a message passing from a feature of a node near $v_s^{(b)}$ to a feature of a node far from $v_s^{(b)}$. This term makes the synthetic feature spread widely compared to observed features. The design goals of the two terms naturally combine through multiplication in Eq. 7. $\xi_{i,b}^*$ is 1 for both $v \in \mathcal{V}_k^{(b)}$ and $\mathrm{x}_s^{(b)}$. However, $\xi_{i,b}^s$ is 1 for $\mathrm{x}_s^{(b)}$ while it is at most $\beta$ for $v \in \mathcal{V}_k^{(b)}$. Therefore, in the second stage diffusion, the synthetic feature has a greater influence than observed features.

To construct a transition matrix, we prepare a row-stochastic matrix by normalizing $\boldsymbol{M}^{(b)}$ to $\overline{\boldsymbol{M}}^{(b)} = (\boldsymbol{D}'^{(b)})^{-1}\boldsymbol{W}^{(b)}$ where $\boldsymbol{D}'^{(b)}$ is a diagonal matrix with $\boldsymbol{D}_{ii}'^{(b)} = \sum_j \boldsymbol{M}_{i,j}$. Then, we replace the first $|\mathcal{V}_{k^*}^{(b)}|$ rows in $\overline{\boldsymbol{M}}$ with one-hot vectors representing $\mathcal{V}_{k^*}^{(b)}$ to preserve $\boldsymbol{x}_{k^*}^{(b)}$ including $\mathrm{x}_s^{(b)}$. By the replacement, we obtain a DSF transition matrix $\widetilde{\boldsymbol{M}}^{(b)}$ as follows:

$$\widetilde{\boldsymbol{M}}^{(b)} = \begin{bmatrix} \boldsymbol{I}_{k^*k^*} & \boldsymbol{0}_{k^*u^*} \\ \overline{\boldsymbol{M}}_{u^*k^*}^{(b)} & \overline{\boldsymbol{M}}_{u^*u^*}^{(b)} \end{bmatrix}, \tag{8}$$

where $\boldsymbol{I}_{k^*k^*} \in \mathbb{R}^{|\mathcal{V}_{k^*}^{(b)}| \times |\mathcal{V}_{k^*}^{(b)}|}$ is an identity matrix and $\boldsymbol{0}_{k^*u^*} \in \mathbb{R}^{|\mathcal{V}_{k^*}^{(b)}| \times |\mathcal{V}_{u^*}^{(b)}|}$ is a zero matrix.

We define diffusion with synthetic features (DSF) by

$$\hat{\boldsymbol{x}}^{(b)}(t) = \widetilde{\boldsymbol{M}}^{(b)}\hat{\boldsymbol{x}}^{(b)}(t-1), \ \ t = 1, \cdots, K;$$
$$\hat{\boldsymbol{x}}^{(b)}(0) = \begin{bmatrix} \boldsymbol{x}_{k^*}^{(b)} \\ \boldsymbol{0}_{u^*} \end{bmatrix}, \tag{9}$$

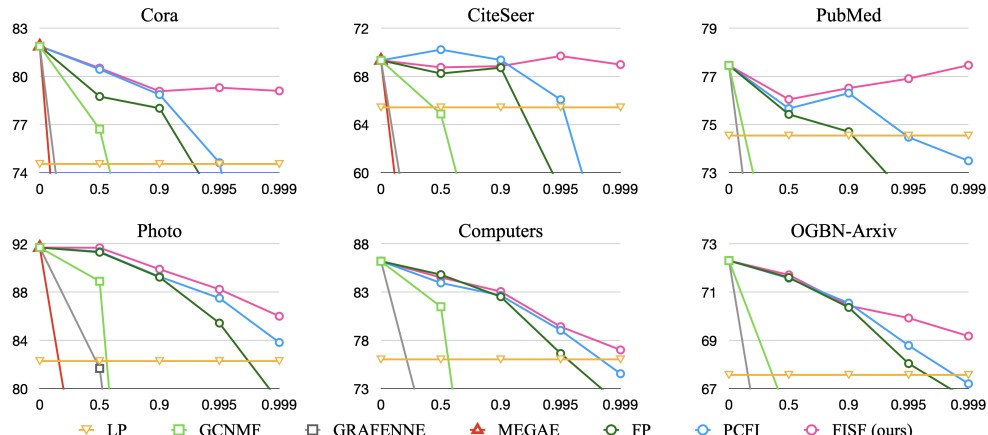

Figure 3: Accuracy (%) on semi-supervised node classification tasks under structural-missing setting with $r_m \in \{0.5, 0.9, 0.995, 0.999\}$. Cases where accuracy cannot be measured due to out-of-memory errors are not included.

where $\hat{\boldsymbol{x}}^{(b)}(t)$ denotes an imputed feature vector after $t$ propagation steps and $\mathbf{0}_{u^*}$ denotes a zero vector of the same length as $|\mathcal{V}_{u^*}^{(b)}|$. As $K \to \infty$, $\hat{\boldsymbol{x}}^{(b)}(K)$ converges (see the proof in Appendix A). With sufficiently large $K$, we approximate the steady state $\lim_{t \to \infty} \hat{\boldsymbol{x}}^{(b)}(t)$ to $\hat{\boldsymbol{x}}^{(b)}(K)$. We perform DSF in the $b$-th channel for all $b \in \mathbb{F}_l$ and obtain $\{\hat{\boldsymbol{x}}^{(b)}(K)\}_{b \in \mathbb{F}_l}$. Since vectors in $\{\hat{\boldsymbol{x}}^{(b)}(K)\}_{b \in \mathbb{F}_l}$ have different ordering from the original one, we restore ordering of all the vectors according to the original order. To construct $\hat{\boldsymbol{X}} \in \mathbb{R}^{N \times F}$, we prepare $\tilde{\boldsymbol{X}} \in \mathbb{R}^{N \times F}$ from the pre-diffusion and replace $\tilde{\boldsymbol{X}}_{:,b}$ for all $b \in \mathbb{F}_l$ with the corresponding vector in $\{\hat{\boldsymbol{x}}^{(b)}(K)\}_{b \in \mathbb{F}_l}$. The feature matrix with the replaced columns is $\hat{\boldsymbol{X}}$, a final output of FISF. $\hat{\boldsymbol{X}}$ is fed to a GNN to perform a given task.

## 5 EXPERIMENTS

We perform comparative evaluation of FISF against state-of-the-art methods on two main graph learing tasks: semi-supervised node classification and link prediction.

### 5.1 DATASETS AND BASELINES

**Datasets.** We conduct experiments on graph datasets from two different domains: citation networks (Cora, CiteSeer, PubMed (Sen et al., 2008), and OGBN-Arxiv (Hu et al., 2020)) and recommendation networks (Photo and Computers (Shchur et al., 2018)) from Amazon. In the citation networks, nodes and edges represent documents and citation links, respectively. In the case of recommendation networks, nodes represent goods and an edge connects two nodes only when the nodes (*i.e.*, products) are frequently bought together. Further information on the datasets is in Appendix D.1.

**Baselines.** We compare FISF with LP (Zhuŕ & GhahramanifH, 2002) and five state-of-the-art methods for graph learning tasks with missing features. (1) LP that does not use any feature propagates partially given labels for semi-supervised node classification. (2) GCNMF (Taguchi et al., 2021) and (3) GRAFENNE (Gupta et al., 2023) are GNN architecture-based methods. (4) MEGAE (Gao et al., 2023) is a reconstruction-based method. (5) FP (Rossi et al., 2022) and (6) PCFI (Um et al., 2022) is diffusion-based methods. Since imputation methods (including MEGAE, FP, PCFI, and FISF) combine with GNNs to perform downstream tasks, we commonly utilize vanilla GCN (Kipf & Welling, 2016a) models for semi-supervised node classification. In link prediction, we commonly utilize graph auto-encoder (GAE) models for the imputation methods.

### 5.2 EXPERIMENTAL SETUP

We follow the missing setting in Um et al. (2022). To evaluate models on graphs containing missing features, we remove a fixed rate (e.g., 90%) of features in the datasets. A missing rate denoted as $r_m$ represents the rate of feature removal. We fill the positions where features are removed with NaN

Table 1: Performance on semi-supervised node classification tasks at $r_m = 0.995$, measured in accuracy (%). Standard deviation errors are given. OOM denotes an out-of-memory error.

**Structural missing**

| Method | CORA | CITESEER | PUBMED | PHOTO | COMPUTERS | OGBN-ARXIV |
|---|---|---|---|---|---|---|
| Full features | $81.87 \pm 1.59$ | $69.32 \pm 0.57$ | $77.45 \pm 2.17$ | $91.69 \pm 0.78$ | $86.19 \pm 0.78$ | $72.30 \pm 0.10$ |
| LP | $74.54 \pm 1.79$ | $65.42 \pm 1.80$ | $71.67 \pm 4.94$ | $82.27 \pm 2.72$ | $76.01 \pm 1.84$ | $67.56 \pm 0.00$ |
| GCNMF | $31.33 \pm 2.73$ | $24.84 \pm 2.44$ | $40.48 \pm 0.53$ | $25.60 \pm 0.17$ | $37.21 \pm 0.08$ | $9.00 \pm 6.27$ |
| GRAFENNE | $20.2 \pm 10.98$ | $17.58 \pm 2.94$ | $33.12 \pm 2.43$ | $21.10 \pm 17.39$ | $16.31 \pm 11.84$ | $13.66 \pm 12.23$ |
| MEGAE | $33.6 \pm 5.04$ | $29.61 \pm 7.12$ | OOM | $68.78 \pm 2.48$ | $51.07 \pm 2.82$ | OOM |
| FP | $71.86 \pm 2.82$ | $58.61 \pm 1.74$ | $71.96 \pm 3.06$ | $85.42 \pm 3.16$ | $76.62 \pm 1.94$ | $68.03 \pm 0.52$ |
| PCFI | $74.62 \pm 1.78$ | $66.06 \pm 3.26$ | $74.47 \pm 2.54$ | $87.49 \pm 1.50$ | $79.02 \pm 1.22$ | $68.78 \pm 0.25$ |
| FISF | $\mathbf{79.29 \pm 1.72}$ | $\mathbf{69.68 \pm 2.47}$ | $\mathbf{76.90 \pm 1.50}$ | $\mathbf{88.22 \pm 0.79}$ | $\mathbf{79.40 \pm 1.11}$ | $\mathbf{69.92 \pm 0.17}$ |

**Uniform missing**

| Method | CORA | CITESEER | PUBMED | PHOTO | COMPUTERS | OGBN-ARXIV |
|---|---|---|---|---|---|---|
| Full features | $81.87 \pm 1.59$ | $69.32 \pm 0.57$ | $77.45 \pm 2.17$ | $91.69 \pm 0.78$ | $86.19 \pm 0.78$ | $72.30 \pm 0.10$ |
| LP | $74.54 \pm 1.79$ | $65.42 \pm 1.80$ | $71.67 \pm 4.94$ | $82.27 \pm 2.72$ | $76.01 \pm 1.84$ | $67.56 \pm 0.00$ |
| GCNMF | $34.01 \pm 8.08$ | $29.71 \pm 5.12$ | $40.08 \pm 0.45$ | $25.59 \pm 0.16$ | $37.20 \pm 0.08$ | $5.86 \pm 0.00$ |
| GRAFENNE | $20.55 \pm 13.65$ | $19.32 \pm 7.42$ | $34.75 \pm 4.26$ | $29.96 \pm 20.92$ | $21.74 \pm 15.94$ | $15.52 \pm 11.70$ |
| MEGAE | $29.77 \pm 0.42$ | $26.10 \pm 3.40$ | OOM | $59.36 \pm 3.01$ | $42.32 \pm 3.17$ | OOM |
| FP | $77.58 \pm 1.98$ | $68.55 \pm 2.33$ | $72.62 \pm 4.18$ | $87.50 \pm 1.49$ | $80.75 \pm 0.70$ | $68.82 \pm 0.07$ |
| PCFI | $78.82 \pm 1.48$ | $68.94 \pm 1.95$ | $76.28 \pm 2.52$ | $88.09 \pm 1.41$ | $81.80 \pm 0.71$ | $69.26 \pm 0.17$ |
| FISF | $\mathbf{79.09 \pm 1.73}$ | $\mathbf{69.52 \pm 1.81}$ | $\mathbf{77.53 \pm 1.28}$ | $\mathbf{88.32 \pm 1.37}$ | $\mathbf{82.12 \pm 0.51}$ | $\mathbf{69.81 \pm 0.16}$ |

values. We remove features in the following two ways: *structural missing* and *uniform missing*. First, in the case of structural missing, we randomly select nodes at a ratio of $r_m$ from entire nodes and remove all the features of the selected nodes. Second, uniform missing removes randomly selected feature values with a ratio of $r_m$ from a feature matrix $\boldsymbol{X}$. We report average performance (e.g., accuracy, ROC AUC, and AP) after five runs of experiments under a fixed setting. Therefore, for each missing way, we randomly generate five different binary masks with the same size of $\boldsymbol{X}$ for each dataset. These masks indicate the locations in $\boldsymbol{X}$ where features are missing.

For semi-supervised node classification tasks, we randomly create five different training/validation/test node splits for all the datasets except for OGBN-Arxiv which has a fixed split according to the specific criteria. For link prediction tasks, we also randomly create five different training/validation/test edge splits of each dataset. OGBN-Arxiv is excluded from the link prediction tasks due to out-of-memory errors. Grid search is employed to tune $\alpha$, $\beta$, and $\gamma$, the three hyper-parameters of FISF. $\alpha$ and $\beta$ are searched within $\{0.1, 0.3, 0.5, 0.7, 0.9\}$. $\gamma$ is chosen from $\{10, 30, 50, 70, 90\}$. For all the methods including FISF, we tune hyper-parameters based on validation sets. We utilize the publicly released code for all the baselines. All models are implemented in Pytorch (Paszke et al., 2019) and Pytorch Geometric (Fey & Lenssen, 2019). Further implementation details including dataset splits, training details, and baseline implementations are provided in Appendix D.2.

## 5.3 SEMI-SUPERVISED NODE CLASSIFICATION RESULTS

To investigate how $r_m$ affects semi-supervised node classification accuracy, we conduct experiments by increasing $r_m$ while keeping all other settings fixed. Figure 3 demonstrates accuracy under structural-missing settings with varying $r_m$. The accuracy of LP remains consistent since LP does not utilize features. For all methods except for LP, the accuracy tends to decrease as $r_m$ increases. While diffusion-based imputation methods outperform the other methods, FP and PCFI suffer performance degradation as $r_m$ increases. However, FISF shows robust performance despite high $r_m$ regardless of the datasets. The performance gain of PCFI tends to increase as $r_m$ increases and the gain is significant when $r_m \in \{0.995, 0.999\}$. Note that FISF using only $0.1\%$ of features (*i.e.*, $r_m = 0.999$) performs similarly to or even outperforms FISF with full features on Cora, CiteSeer, and PubMed. Moreover, for $r_m \in \{0.5, 0.9\}$, FISF improves the classification accuracy under most missing settings.

We then conduct experiments to investigate how semi-supervised node classification accuracy varies depending on the missing ways (structural and uniform missing) at the same $r_m = 0.995$. Table 1

Table 2: Performance on link prediction tasks at $r_m = 0.995$, measured in ROC AUC score (%). Standard deviation errors are given. The best result is highlighted in bold and underlined, while the second-best result is highlighted only in bold. OOM denotes an out-of-memory error.

*Structural missing*

| Method | CORA | CITESEER | PUBMED | PHOTO | COMPUTERS |
|---|---|---|---|---|---|
| Full features | $92.20 \pm 0.96$ | $90.55 \pm 1.36$ | $96.41 \pm 0.25$ | $95.70 \pm 0.32$ | $93.71 \pm 0.65$ |
| GCNMF | $67.44 \pm 0.45$ | $68.34 \pm 1.79$ | $\mathbf{87.20 \pm 0.28}$ | $81.00 \pm 0.25$ | $82.92 \pm 0.19$ |
| GRAFENNE | $53.79 \pm 5.26$ | $62.96 \pm 13.82$ | $60.11 \pm 6.10$ | $66.44 \pm 1.74$ | $67.23 \pm 1.71$ |
| MEGAE | $67.13 \pm 0.75$ | $69.34 \pm 2.46$ | OOM | $86.53 \pm 1.97$ | $84.89 \pm 1.77$ |
| FP | $83.85 \pm 1.32$ | $79.83 \pm 2.18$ | $78.54 \pm 1.13$ | $94.25 \pm 0.58$ | $91.27 \pm 0.71$ |
| PCFI | $86.75 \pm 0.84$ | $79.38 \pm 1.81$ | $82.49 \pm 0.82$ | $\mathbf{96.65 \pm 0.25}$ | $94.54 \pm 0.37$ |
| FISF | $\mathbf{\underline{87.26 \pm 1.44}}$ | $\mathbf{84.12 \pm 1.17}$ | $83.19 \pm 0.78$ | $95.86 \pm 0.21$ | $\mathbf{94.70 \pm 0.30}$ |
| FISF+NIP | $\mathbf{\underline{87.16 \pm 1.46}}$ | $\mathbf{\underline{84.20 \pm 1.70}}$ | $83.28 \pm 0.42$ | $96.35 \pm 0.18$ | $\mathbf{\underline{95.29 \pm 0.32}}$ |

*Uniform missing*

| Method | CORA | CITESEER | PUBMED | PHOTO | COMPUTERS |
|---|---|---|---|---|---|
| Full features | $92.20 \pm 0.96$ | $90.55 \pm 1.36$ | $96.41 \pm 0.25$ | $95.70 \pm 0.32$ | $93.71 \pm 0.65$ |
| GCNMF | $63.46 \pm 1.04$ | $63.50 \pm 3.40$ | $81.73 \pm 3.13$ | $80.98 \pm 0.17$ | $82.95 \pm 0.11$ |
| GRAFENNE | $68.49 \pm 17.00$ | $61.38 \pm 13.53$ | $65.74 \pm 11.32$ | $68.53 \pm 6.57$ | $70.16 \pm 4.12$ |
| MEGAE | $65.86 \pm 1.22$ | $62.21 \pm 3.18$ | OOM | $84.25 \pm 1.35$ | $84.95 \pm 2.20$ |
| FP | $86.79 \pm 1.36$ | $81.55 \pm 2.30$ | $76.87 \pm 2.89$ | $95.96 \pm 0.17$ | $94.10 \pm 0.33$ |
| PCFI | $87.35 \pm 1.28$ | $82.33 \pm 1.88$ | $84.68 \pm 0.75$ | $\mathbf{\underline{97.05 \pm 0.16}}$ | $\mathbf{95.62 \pm 0.24}$ |
| FISF | $\mathbf{87.44 \pm 0.80}$ | $\mathbf{\underline{83.45 \pm 2.53}}$ | $\mathbf{\underline{85.33 \pm 0.47}}$ | $96.64 \pm 0.18$ | $95.13 \pm 0.35$ |
| FISF+NIP | $\mathbf{\underline{87.70 \pm 0.77}}$ | $\mathbf{82.53 \pm 1.94}$ | $\mathbf{85.32 \pm 0.48}$ | $96.67 \pm 0.21$ | $\mathbf{\underline{96.09 \pm 0.24}}$ |

summarizes the classification accuracy of FISF and the other methods. While most nodes have some observed features in uniform-missing settings, $(1-r_m)$ of nodes do not have observed features at all in structural-missing settings. Therefore, the performance of methods tends to be better in uniform-missing settings than in structural-missing settings. For both missing ways, FISF outperforms the state-of-the-art methods across all the datasets.

## 5.4 LINK PREDICTION RESULTS

Table 2 summarizes the ROC AUC score on link prediction tasks at $r_m = 0.995$. (The AP comparison results are in Appendix E.1.) NIP denotes node-wise inter-channel propagation included in PCFI (Um et al., 2022), which refines an output matrix from channel-wise diffusion. Since NIP is effective in link prediction tasks, we demonstrate the ROC AUC score of FISF and FISF+NIP (FISF followed by NIP). FISF and FISF+NIP achieve state-of-the-art performance in three and four settings, respectively, out of 10 settings. Even in the remaining three settings, FISF+NIP still demonstrates the second-best scores which are comparable with the best scores. That is, FISF and FISF+NIP achieve strong performance across all five datasets regardless of missing ways. As highlighted scores in Table 2 shows, FISF demonstrates its effectiveness on link prediction tasks with missing features.

## 6 CONCLUSION

In this paper, we propose a novel scheme called Feature Imputation with Synthetic Features (FISF) for graph feature imputation. FISF consists of two diffusion stages: pre-diffusion and diffusion with synthetic features (DSF). The pre-diffusion stage outputs a pre-imputed feature matrix and then feature variances of channels are calculated using the pre-imputed matrix. After generating synthetic features in low-variance channels, DSF spreads these synthetic features widely by utilizing distance encoding, resulting in the final imputed feature matrix. Through extensive experiments on both semi-supervised node classification and link prediction, we demonstrate that FISF outperforms existing methods handling graph learning tasks with missing features. FISF can extend its applicability to hypergraphs and heterogeneous through clique expansion and meta-paths, repectively. We leave the extension of FISF as future work.

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

## A    PROOF OF CONVERGENCE OF DIFFUSION STAGES

Our FISF consists of two diffusion stages: pre-diffusion and DSF. Both stages utilize row stochastic transition matrices for diffusion. We prove the convergence of the two diffusion stages as follows.

**Proposition 1.** *The pre-diffusion transition matrix for the $a$-th channel is defined by*

$$\widetilde{\boldsymbol{W}}^{(a)} = \begin{bmatrix} \boldsymbol{I}_{kk} & \boldsymbol{0}_{ku} \\ \boldsymbol{W}^{(a)}_{uk} & \boldsymbol{W}^{(a)}_{uu} \end{bmatrix},$$

*where $\widetilde{\boldsymbol{W}}^{(a)}$ is a row-stochastic. Using $\widetilde{\boldsymbol{W}}^{(a)}$, the pre-diffusion in the $a$-th channel is defined by*

$$\tilde{\boldsymbol{x}}^{(a)}(t) = \widetilde{\boldsymbol{W}}^{(a)} \tilde{\boldsymbol{x}}^{(a)}(t-1), \;\; t = 1, \cdots, K;$$

$$\tilde{\boldsymbol{x}}^{(a)}(0) = \begin{bmatrix} \boldsymbol{x}^{(a)}_k \\ \boldsymbol{0}_u \end{bmatrix},$$

*Then, $\lim\limits_{K \to \infty} \tilde{\boldsymbol{x}}^{(a)}(K)$ converges.*

The proof of Propostion 1 refers to Um et al. (2022). After we establish the convergence of pre-diffusion, we demonstrate that this proof extends to cover the convergence of DSF. To start, we introduce two lemmas.

**Lemma 1.** $\overline{\boldsymbol{W}}^{(a)}$ *is the row-stochastic matrix calculated by* $\overline{\boldsymbol{W}}^{(a)} = (\boldsymbol{D}^{(a)})^{-1} \boldsymbol{W}^{(a)}$ *where* $\boldsymbol{D}^{(a)}$ *is a diagonal matrix that has diagonal entities* $\boldsymbol{D}^{(a)}_{ii} = \sum_j \boldsymbol{W}_{i,j}$. $\overline{\boldsymbol{W}}^{(a)}_{uu}$ *is the* $|\hat{\boldsymbol{x}}^{(a)}_u| \times |\hat{\boldsymbol{x}}^{(a)}_u|$ *bottom-right submatrix of* $\overline{\boldsymbol{W}}^{(a)}$ *and let* $\rho(\cdot)$ *denote spectral radius. Then,* $\rho(\overline{\boldsymbol{W}}^{(a)}_{uu}) < 1$.

*Proof.* Consider $\overline{\boldsymbol{W}}^{(a)}_{uu0} \in \mathbb{R}^{N \times N}$, where the bottom right submatrix is denoted as $\overline{\boldsymbol{W}}^{(a)}_{uu}$ and all the other elements are zero. That is,

$$\overline{\boldsymbol{W}}^{(a)}_{uu0} = \begin{bmatrix} \boldsymbol{0}_{kk} & \boldsymbol{0}_{ku} \\ \boldsymbol{0}_{uk} & \overline{\boldsymbol{W}}^{(a)}_{uu} \end{bmatrix}$$

where $\boldsymbol{0}_{kk} \in \{0\}^{|\hat{\boldsymbol{x}}^{(a)}_k| \times |\hat{\boldsymbol{x}}^{(a)}_k|}$, $\boldsymbol{0}_{ku} \in \{0\}^{|\hat{\boldsymbol{x}}^{(a)}_k| \times |\hat{\boldsymbol{x}}^{(a)}_u|}$, and $\boldsymbol{0}_{uk} \in \{0\}^{|\hat{\boldsymbol{x}}^{(a)}_u| \times |\hat{\boldsymbol{x}}^{(a)}_k|}$. Given that $\overline{\boldsymbol{W}}^{(a)}$ represents the weighted adjacency matrix of the connected graph $\mathcal{G}$, $\overline{\boldsymbol{W}}^{(a)}_{uu0} \leq \overline{\boldsymbol{W}}^{(a)}$ element-wisely and $\overline{\boldsymbol{W}}^{(a)}_{uu0} \neq \overline{\boldsymbol{W}}^{(a)}$. Furthermore, considering that $\overline{\boldsymbol{W}}^{(a)}_{uu0} + \overline{\boldsymbol{W}}^{(a)}$ constitutes the weighted adjacency matrix of a strongly connected graph, we can conclude that $\overline{\boldsymbol{W}}^{(a)}_{uu0} + \overline{\boldsymbol{W}}^{(a)}$ is irreducible based on Theorem 2.2.7 in Berman & Plemmons (1994). Consequently, applying Corollary 2.1.5 in Berman & Plemmons (1994), $\rho(\overline{\boldsymbol{W}}^{(a)}_{uu0}) < \rho(\overline{\boldsymbol{W}}^{(a)})$. Since the spectral radius of a stochastic matrix is one according to Theorem 2.5.3 in Berman & Plemmons (1994), we have $\rho(\overline{\boldsymbol{W}}^{(a)}) = 1$. Moreover, since both $\overline{\boldsymbol{W}}^{(a)}_{uu0}$ and $\overline{\boldsymbol{W}}^{(a)}_{uu}$ share the same non-zero eigenvalues, it follows that $\rho(\overline{\boldsymbol{W}}^{(a)}_{uu0}) = \rho(\overline{\boldsymbol{W}}^{(a)}_{uu})$. Ultimately, this leads to the conclusion that $\rho(\overline{\boldsymbol{W}}^{(a)}_{uu}) = \rho(\overline{\boldsymbol{W}}^{(a)}_{uu0}) < \rho(\overline{\boldsymbol{W}}^{(a)}) = 1$. □

**Lemma 2.** $\boldsymbol{I}_{uu} - \overline{\boldsymbol{W}}^{(a)}_{uu}$ *is invertible where* $\boldsymbol{I}_{uu}$ *is the* $|\hat{\boldsymbol{x}}^{(a)}_u| \times |\hat{\boldsymbol{x}}^{(a)}_u|$ *identity matrix.*

*Proof.* Since 1 is not an eigenvalue of $\overline{\boldsymbol{W}}^{(a)}_{uu}$ by Lemma 1, 0 is not an eigenvlaue of $\boldsymbol{I}_{uu} - \overline{\boldsymbol{W}}^{(a)}_{uu}$. Thus $\boldsymbol{I}_{uu} - \overline{\boldsymbol{W}}^{(a)}_{uu}$ is invertible. □

We now prove Propostion 1 as follows.

*Proof.* Unfolding the recurrence relation gives us

$$\hat{\boldsymbol{x}}^{(a)}(t) = \begin{bmatrix} \hat{\boldsymbol{x}}^{(a)}_k(t) \\ \hat{\boldsymbol{x}}^{(a)}_u(t) \end{bmatrix} = \begin{bmatrix} \boldsymbol{I}_{kk} & \boldsymbol{0}_{ku} \\ \overline{\boldsymbol{W}}^{(a)}_{uk} & \overline{\boldsymbol{W}}^{(a)}_{uu} \end{bmatrix} \begin{bmatrix} \hat{\boldsymbol{x}}^{(a)}_k(t-1) \\ \hat{\boldsymbol{x}}^{(a)}_u(t-1) \end{bmatrix} = \begin{bmatrix} \hat{\boldsymbol{x}}^{(a)}_k(t-1) \\ \overline{\boldsymbol{W}}^{(a)}_{uk} \hat{\boldsymbol{x}}^{(a)}_k(t-1) + \overline{\boldsymbol{W}}^{(a)}_{uu} \hat{\boldsymbol{x}}^{(a)}_u(t-1) \end{bmatrix}.$$

Since $\hat{\boldsymbol{x}}_k^{(a)}(t) = \hat{\boldsymbol{x}}_k^{(a)}(t-1)$ in the first $|\hat{\boldsymbol{x}}_k^{(a)}|$ rows, it follows that $\hat{\boldsymbol{x}}_k^{(a)}(K) = \ldots = \hat{\boldsymbol{x}}_k^{(a)}$. That is, $\hat{\boldsymbol{x}}_k^{(a)}(K)$ retains the values of $\boldsymbol{x}_k^{(a)}$. Therefore, $\lim_{K \to \infty} \hat{\boldsymbol{x}}_k^{(a)}(K)$ converges to $\boldsymbol{x}_k^{(a)}$.

Now, we focus solely on the convergence of $\lim_{K \to \infty} \hat{\boldsymbol{x}}_u^{(a)}(K)$. When we unroll the recursion for the last $|\hat{\boldsymbol{x}}_u^{(a)}|$ rows,

$$
\begin{aligned}
\hat{\boldsymbol{x}}_u^{(a)}(K) &= \overline{\boldsymbol{W}}_{uk}^{(a)} \boldsymbol{x}_k^{(a)} + \overline{\boldsymbol{W}}_{uu}^{(a)} \hat{\boldsymbol{x}}_u^{(a)}(K-1) \\
&= \overline{\boldsymbol{W}}_{uk}^{(a)} \boldsymbol{x}_k^{(a)} + \overline{\boldsymbol{W}}_{uu}^{(a)} (\overline{\boldsymbol{W}}_{uk}^{(a)} \boldsymbol{x}_k^{(a)} + \overline{\boldsymbol{W}}_{uu}^{(a)} \hat{\boldsymbol{x}}_u^{(a)}(K-2)) \\
&= \ldots \\
&= \Big( \sum_{t=0}^{K-1} (\overline{\boldsymbol{W}}_{uu}^{(a)})^t \Big) \overline{\boldsymbol{W}}_{uk}^{(a)} \boldsymbol{x}_k^{(a)} + (\overline{\boldsymbol{W}}_{uu}^{(a)})^K \hat{\boldsymbol{x}}_u^{(a)}(0)
\end{aligned}
$$

By Lemma 1, $\lim_{K \to \infty} (\overline{\boldsymbol{W}}_{uu}^{(a)})^K = 0$. Therefore, $\lim_{K \to \infty} (\overline{\boldsymbol{W}}_{uu}^{(a)})^K \hat{\boldsymbol{x}}_u^{(a)}(0) = 0$, regardless of the initial state for $\hat{\boldsymbol{x}}_u^{(a)}(0)$. (we replace $\hat{\boldsymbol{x}}_u^{(a)}(0)$ with a zero column vector for simplicity.) Hence, our focus shifts to $\lim_{K \to \infty} (\sum_{t=0}^{K-1} (\overline{\boldsymbol{W}}_{uu}^{(a)})^t) \overline{\boldsymbol{W}}_{uk}^{(a)} \boldsymbol{x}_k^{(a)}$.

Given that Lemma 1 establishes $\rho(\overline{\boldsymbol{W}}_{uu}^{(a)}) < 1$, and Lemma 2 affirms the invertibility of $(\boldsymbol{I}_{uu} - \overline{\boldsymbol{W}}_{uu}^{(a)})^{-1}$, the geometric series converges as follows

$$
\lim_{K \to \infty} \hat{\boldsymbol{x}}_u^{(a)}(K) = \lim_{K \to \infty} \Big( \sum_{t=0}^{K-1} (\overline{\boldsymbol{W}}_{uu}^{(a)})^t \Big) \overline{\boldsymbol{W}}_{uk}^{(a)} \boldsymbol{x}_k^{(a)} = (\boldsymbol{I}_{uu} - \overline{\boldsymbol{W}}_{uu}^{(a)})^{-1} \overline{\boldsymbol{W}}_{uk}^{(a)} \boldsymbol{x}_k^{(a)}.
$$

In conclusion, the recursion in the pre-diffusion converges. $\qquad\square$

In the case of DSF, the DSF transition matrix $\widetilde{\boldsymbol{M}}^{(b)}$ in Eq. 8 is also row stochastic. The distinction between $\widetilde{\boldsymbol{W}}^{(a)}$ and $\widetilde{\boldsymbol{M}}^{(b)}$ lies solely in the number of channels where diffusion is performed and the sizes of each sub-matrix. Therefore, the convergence of the DSF can also be established through the proof of Proposition 1.

## B  PROOF OF THE PROPOSITION IN SEC 4.3

We refer to the proposition in Section 4.3 as Proposition 2.

**Proposition 2.** *In pre-diffusion (channel-wise inter-node diffusion (Um et al., 2022)), when all given known features in the $a$-th channel (i.e., elements in $\boldsymbol{x}_k^{(a)}$) have the same value $c$, $\lim_{t \to \infty} \tilde{\boldsymbol{x}}^{(a)}(t)$ becomes a vector where entire elements are equal to $c$.*

*Proof.* In accordance with the given assumption, entire elements in $\boldsymbol{x}_k^{(a)}$ have the value of $c$. Here, we can initialize $\hat{\boldsymbol{x}}^{(a)}(0)$ with the same values as $c$. According to the proof of Proposition 1, $\lim_{K \to \infty} \hat{\boldsymbol{x}}_u^{(a)}(K) = (\boldsymbol{I}_{uu} - \overline{\boldsymbol{W}}_{uu}^{(a)})^{-1} \overline{\boldsymbol{W}}_{uk}^{(a)} \boldsymbol{x}_k^{(a)}$ and $\hat{\boldsymbol{x}}_k^{(a)}(K) = \boldsymbol{x}_k^{(a)}$. This means that initializing $\hat{\boldsymbol{x}}^{(a)}(0)$ with the values of $c$ does not affect the final output, $\lim_{K \to \infty} \hat{\boldsymbol{x}}^{(a)}(K)$. Formally, pre-diffusion of which steady state is the same as that of Eq. 5 can be expressed as follows:

$$
\begin{aligned}
\tilde{\boldsymbol{x}}^{(a)}(t) &= \widetilde{\boldsymbol{W}}^{(a)} \tilde{\boldsymbol{x}}^{(a)}(t-1), \quad t = 1, \cdots, K; \\
\tilde{\boldsymbol{x}}^{(a)}(0) &= \begin{bmatrix} \boldsymbol{c}_k \\ \boldsymbol{c}_u \end{bmatrix},
\end{aligned}
\tag{10}
$$

where $\boldsymbol{c}_k$ and $\boldsymbol{c}_u$ are column vectors with lengths of $|\mathcal{V}_k^{(a)}|$ and $|\mathcal{V}_u^{(a)}|$, respectively, filled only with the value $c$.

Since $\widetilde{\boldsymbol{W}}^{(a)}$ is row stochastic, $\sum_{j=0}^{K-1} \widetilde{\boldsymbol{W}}_{i,j}^{(a)} = 1$ for all $i \in \{1, \ldots, N\}$. Therefore, in Eq. 10, the $i$-th element in $\tilde{\boldsymbol{x}}^{(a)}(1)$ is calculated as $\sum_{j=0}^{K-1} \widetilde{\boldsymbol{W}}_{i,j}^{(a)} \cdot c = c \cdot \sum_{j=0}^{K-1} \widetilde{\boldsymbol{W}}_{i,j}^{(a)} = c$ for all $i \in \{1, \ldots, N\}$. That is, $\tilde{\boldsymbol{x}}^{(a)}(1)$ is filled only with the value $c$, which is the same as $\tilde{\boldsymbol{x}}^{(a)}(0)$. Thus, even if this recursion repeats, $\tilde{\boldsymbol{x}}^{(a)}(t)$ remains the same as $\begin{bmatrix} \boldsymbol{c}_k \\ \boldsymbol{c}_u \end{bmatrix}$, which results in $\lim_{t \to \infty} \tilde{\boldsymbol{x}}^{(a)}(t) = \begin{bmatrix} \boldsymbol{c}_k \\ \boldsymbol{c}_u \end{bmatrix}$ where entire elements are equal to $c$.  □

## C  ADDITIONAL EXPERIMENTS

### C.1  ABLATION STUDY

Table 3: Ablation study of FISF. SS node classification denotes semi-supervised node classification. # denotes the number of synthetic features injected into a low-variance channel. * denotes the optimal hyper-parameter at the setting.

| Task |  |  | SS node classification | Link prediction | |
|---|---|---|---|---|---|
| Dataset |  |  | Cora | CiteSeer | |
| # | $\beta$ | $\gamma$ | ACC | AUC | AP |
| 1 | 1 | 0 | $74.62 \pm 1.78$ | $79.38 \pm 1.81$ | $82.98 \pm 0.86$ |
| 1 | 1 | 100 | $78.50 \pm 1.91$ | $83.63 \pm 1.69$ | $85.42 \pm 1.79$ |
| 1 | 1 | * | $78.52 \pm 1.94$ | $83.46 \pm 1.84$ | $85.32 \pm 1.59$ |
| 1 | * | 100 | $78.78 \pm 1.51$ | $58.67 \pm 13.44$ | $60.27 \pm 14.40$ |
| 2 | * | * | $78.88 \pm 1.91$ | $82.11 \pm 2.43$ | $83.61 \pm 2.50$ |
| 1 | * | * | $\mathbf{79.29 \pm 1.72}$ | $\mathbf{84.12 \pm 1.17}$ | $\mathbf{85.85 \pm 1.38}$ |

We conduct an ablation study to investigate the effectiveness of the elements in FISF. We perform both semi-supervised node classification and link prediction. For ablation study on semi-supervised node classification, we conduct experiments on Cora under a structural-missing setting with $r_m = 0.995$. For link prediction, we utilize CiteSeer under a structural-missing setting with $r_m = 0.995$. $\beta$ takes a role in spreading synthetic features widely and $\gamma$ implies the ratio of selected low-variance channels to diffuse with synthetic features. Table 3 demonstrates the results of the ablation study. The results show that the performance gain by introducing synthetic features (*i.e.*, $\gamma \neq 0$) is significant. The optimal $\beta$ and the optimal $\gamma$ synergistically enhance the performance, resulting in considerable improvements. The bottom two rows in Table 3 demonstrate that injecting two synthetic features into row-variance channels leads to degradation in performance. This shows the validity of injecting a single synthetic feature into a low-variance channel.

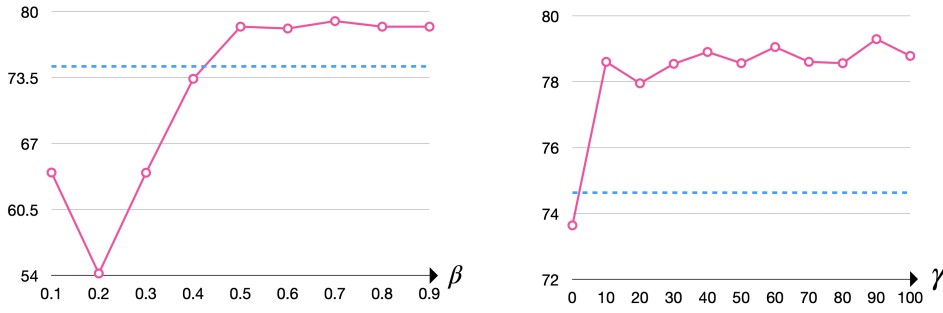

Figure 4: Semi-supervised node classification accuracy with different $\beta$ and $\gamma$. The blue dashed lines indicate existing state-of-the-art performance.

We further analyze the effect of $\beta$ and $\gamma$ on Cora under a structural missing settings with $r_m = 0.995$. Figure 4 shows the accuracy of FISF models with different $\beta$ and $\gamma$. A small $\beta$ results in the performance degradation. This is because too small $\beta$ assigns excessive influence to synthetic features, which hinders the spread of known features. We can observe significant performance improvement by introducing with small $\gamma$.

## C.2 HYPERPARAMETER SEARCH FOR FISF

Despite the outperforming performance of FISF, conducting a hyperparameter search for FISF with three hyperparameters ($\alpha$, $\beta$, and $\gamma$) can be burdensome in certain situations. However, both $\alpha$ and $\beta$ ($0 < \alpha, \beta < 1$) play a shared role in a base of distance during calculating PC (*i.e.* $\xi_{i,b}^* = \alpha^{S_{i,b}^*}$ and $\xi_{i,a}^s = \beta^{S_{i,a}^s}$). Thus we can combine them into one, i.e., $\alpha = \beta$. By doing this, the search complexity can be reduced from $5^3$ to $5^2$ without the performance degradation by setting five search points for each hyperparameter. Table 4 and Table 5 show that the FISF* with the light search does not degrade performance on semi-supervised node classification and link prediction. The version with the light search requires from 20 minutes to 10 hours depending on the datasets, therefore this burden is manageable for practical usage of FISF.

Table 4: Performance on semi-supervised node classification tasks at $r_m = 0.995$, measured in accuracy (%).

|  | | | | | | | |
| --- | --- | --- | --- | --- | --- | --- | --- |
| | | | ***Structural missing*** | | | | |
| Method | CORA | CITESEER | PUBMED | PHOTO | COMPUTERS | OGBN-ARXIV | Average |
| FISF | $79.29 \pm 1.72$ | $69.68 \pm 2.47$ | $76.90 \pm 1.50$ | $88.22 \pm 0.79$ | $79.40 \pm 1.11$ | $69.92 \pm 0.17$ | 77.24 |
| FISF* | $78.68 \pm 1.72$ | $69.68 \pm 2.47$ | $76.74 \pm 1.84$ | $88.22 \pm 0.79$ | $79.40 \pm 1.11$ | $69.92 \pm 0.17$ | 77.11 |

|  | | | | | | | |
| --- | --- | --- | --- | --- | --- | --- | --- |
| | | | ***Uniform missing*** | | | | |
| Method | CORA | CITESEER | PUBMED | PHOTO | COMPUTERS | OGBN-ARXIV | Average |
| FISF | $79.09 \pm 1.73$ | $69.52 \pm 1.81$ | $77.53 \pm 1.28$ | $88.32 \pm 1.37$ | $82.12 \pm 0.51$ | $69.81 \pm 0.16$ | 77.73 |
| FISF* | $79.09 \pm 1.73$ | $69.52 \pm 1.81$ | $76.89 \pm 2.01$ | $88.32 \pm 1.37$ | $81.56 \pm 0.47$ | $69.81 \pm 0.16$ | 77.53 |

Table 5: Performance on link prediction tasks at $r_m = 0.995$, measured in ROC AUC score (%).

|  | | | | | |
| --- | --- | --- | --- | --- | --- |
| | | ***Structural missing*** | | | |
| Method | CORA | CITESEER | PUBMED | PHOTO | COMPUTERS | Average |
| FISF | $87.26 \pm 1.44$ | $84.12 \pm 1.17$ | $83.19 \pm 0.78$ | $95.86 \pm 0.21$ | $94.70 \pm 0.30$ | 89.03 |
| FISF* | $86.80 \pm 1.27$ | $84.12 \pm 1.17$ | $82.46 \pm 0.94$ | $95.76 \pm 0.33$ | $94.39 \pm 0.82$ | 88.70 |

|  | | | | | |
| --- | --- | --- | --- | --- | --- |
| | | ***Uniform missing*** | | | |
| Method | CORA | CITESEER | PUBMED | PHOTO | COMPUTERS | Average |
| FISF | $87.44 \pm 0.80$ | $83.45 \pm 2.53$ | $85.33 \pm 0.47$ | $96.64 \pm 0.18$ | $95.13 \pm 0.35$ | 89.60 |
| FISF* | $87.56 \pm 1.29$ | $81.15 \pm 1.17$ | $82.46 \pm 0.69$ | $95.68 \pm 0.42$ | $94.94 \pm 0.27$ | 88.36 |

## C.3 CONTRIBUTION OF LOW-VARIANCE CHANNELS IN DOWNSTREAM TASKS

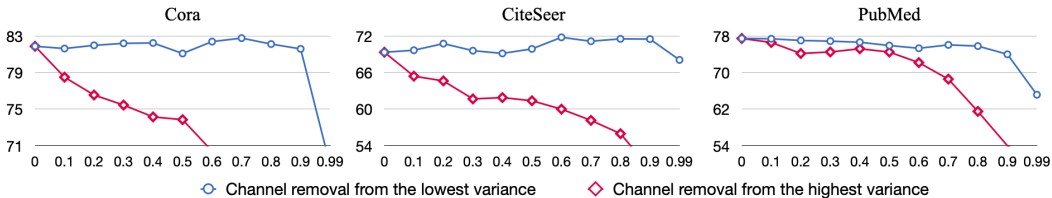

Figure 5: Accuracy (%) on semi-supervised node classification tasks while increasing the proportion of excluded channels from the original feature matrix.

In order to experimentally confirm little contribution of low-variance channels in downstream tasks, we compare performance by excluding partial channels from the original feature matrix using two

different ways. The first way (red lines in Figure 5 and Figure 6) is excluding channels in descending order of variance, starting from the highest, based on a fixed proportion. Then, as the second way (blue lines), we exclude channels from the lowest variance in ascending order, *i.e.*, the low-variance channels are removed first.

Figure 5 demonstrates the results on semi-supervised node classification tasks. Since a low-variance channel contains nearly identical values that do not aid in distinguishing nodes, the classification accuracy denoted by blue lines persists despite an increasing removal proportion of low-variance channels. However, cases of channel removal from the highest variance suffer significant performance degradation even with low proportion of channel removal.

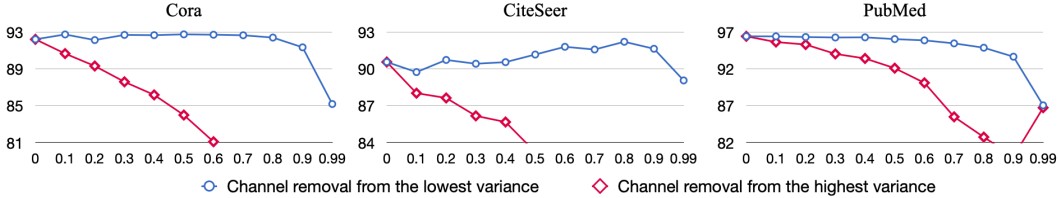

Figure 6: ROC AUC score (%) on link prediction tasks while increasing the proportion of excluded channels from the original feature matrix.

As shown in Figure 6, little contribution of low-variance channels is also evident in link prediction tasks. Since identical representations among nodes results in consistent representations across node pairs, low-variance channels also contribute very little to performance in link prediction tasks.

## C.4 DISCUSSION ON SMOOTHNESS

Table 6: $\log(E_D)$ of imputed features under a structural-missing setting with $r_m = 0.995$, where $E_D$ is the Dirichlet energy. Original denotes original given features.

| Missing way | Structural | | | Uniform | | |
|---|---|---|---|---|---|---|
| Method ↓ | CORA | CITESEER | PUBMED | CORA | CITESEER | PUBMED |
| Original | 4.36 | 4.49 | 3.11 | 4.36 | 4.49 | 3.11 |
| FP | 1.90 | 1.94 | 0.798 | 1.89 | 1.91 | 0.805 |
| PCFI | 3.14 | 2.59 | 1.49 | 2.52 | 2.64 | 1.43 |
| FISF (Ours) | 3.25 | 2.92 | 4.15 | 2.69 | 2.70 | 4.34 |

We generate a synthetic feature in a low-variance channel in order to make features in that channel distinctive across nodes. To investigate smoothness (feature homophily), we compare the smoothness of output features obtained through imputation methods. For this comparison, we employ Dirichlet energy, a representative criterion for measuring smoothness on a graph. As shown in Table 6, FP displays the lowest Dirichlet energy among the imputation methods. In contrast, FISF makes Dirichlet energy of the imputed features similar to that of the original features. Note that our FISF shows the highest Dirichlet energy (distinctiveness) among the methods. Through the outperforming performance of FISF over the existing methods, we can confirm that features with low dirichlet energy (high feature homophily) does not always ensure good performance in downstream tasks while smoothness is an inductive bias of GNNs.

Table 7: Average cosine similarity of imputed features by FISF, under a structural-missing setting with $r_m = 0.995$.

| Dataset | Inter-class | Intra-class | | | | | | | | Ratio |
|---|---|---|---|---|---|---|---|---|---|---|
| | | class 1 | class 2 | class 3 | class 4 | class 5 | class 6 | class 7 | Average | |
| Cora | 0.760 | 0.858 | 0.902 | 0.902 | 0.844 | 0.691 | 0.826 | 0.870 | 0.842 | 1.11 |
| CiteSeer | 0.279 | 0.267 | 0.341 | 0.636 | 0.282 | 0.513 | 0.380 | - | 0.403 | 1.45 |
| PubMed | 0.871 | 0.893 | 0.936 | 0.880 | - | - | - | - | 0.903 | 1.04 |

Table 8: Average cosine similarity of original features.

| Dataset | Inter-class | Intra-class | | | | | | | | Ratio |
|---------|-------------|---------|---------|---------|---------|---------|---------|---------|---------|-------|
| | | class 1 | class 2 | class 3 | class 4 | class 5 | class 6 | class 7 | Average | |
| Cora | 0.0578 | 0.841 | 0.113 | 0.0896 | 0.683 | 0.0690 | 0.0853 | 0.109 | 0.0883 | 1.53 |
| CiteSeer | 0.0470 | 0.655 | 0.0601 | 0.0617 | 0.0650 | 0.762 | 0.0581 | - | 0.0644 | 1.37 |
| PubMed | 0.0719 | 0.112 | 0.937 | 0.0779 | - | - | - | - | 0.0946 | 1.32 |

To investigate smoothness within classes, we conduct further experiments. Table 7 demonstrates the intra-class cosine similarity calculated from imputed features by FISF. Ratio denotes average similarity/inter-class similarity. If Ratio is greater than 1, inter-class similarity becomes less than the average intra-class similarity, which means the feature is distinctive enough for classification of node features.

Table 8 shows the intra-class cosine similarity calculated from original features. The results indicate that original features also have values of Ratio greater than 1 across the datasets. This means that the datasets also originally have higher intra-class feature similarity compared to inter-class feature similarity. Despite the introduction of synthetic features during diffusion, as shown in Table 7, we can observe that imputed features by our scheme consistently maintains higher intra-class feature similarity than inter-class feature similarity.

We also perform qualitative analysis on imputed features and deep features to compare imputation methods. The qualitative analysis is conducted in structural missing settings with $r_m = 0.995$. Figure 7 and Figure 8 demonstrates the t-SNE plots visualizing imputed features and deep features, respectively. FISF provides clearer cluster structures for both imputed features and deep features than the other imputation methods.

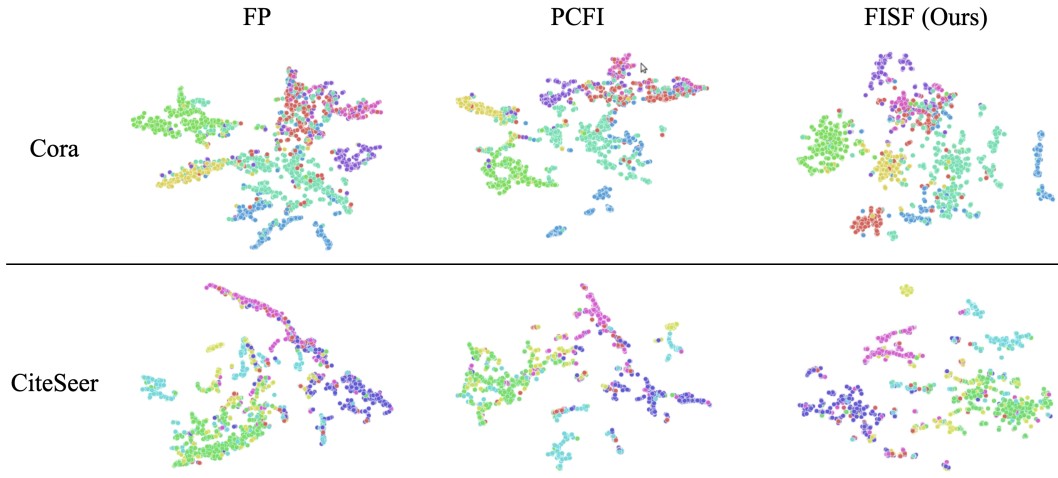

Figure 7: t-SNE plot visualizing imputed features.

FP                              PCFI                           FISF (Ours)

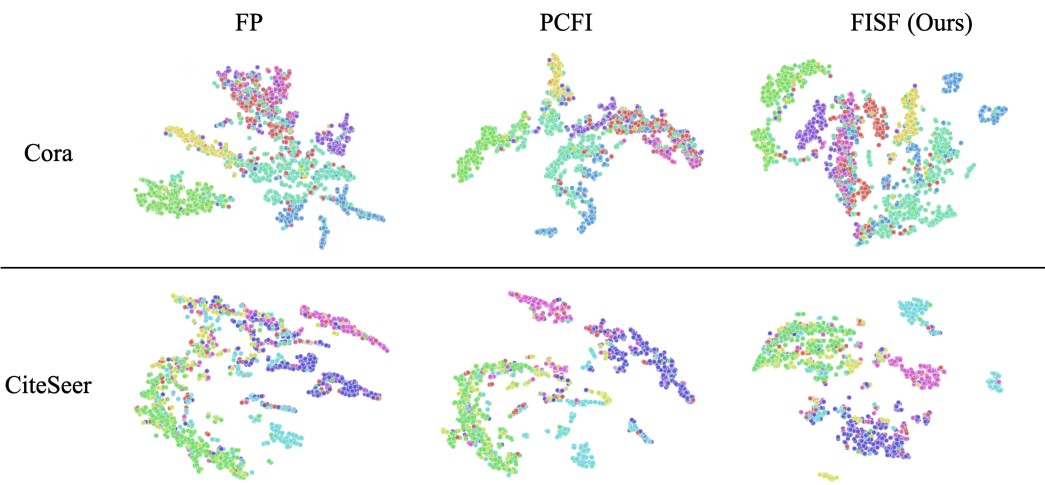

Figure 8: t-SNE plot visualizing deep features in GCN.

## C.5  FASTFISF

Table 9: Performance on semi-supervised node classification tasks at $r_m = 0.995$, measured in accuracy (%). FastFISF denotes FISF using FP instead of PCFI for pre-diffusion.

| | | | Structural missing | | | | |
|---|---|---|---|---|---|---|---|
| Method | CORA | CITESEER | PUBMED | PHOTO | COMPUTERS | OGBN-ARXIV | Average |
| FISF | $79.29 \pm 1.72$ | $69.68 \pm 2.47$ | $76.90 \pm 1.50$ | $88.22 \pm 0.79$ | $79.40 \pm 1.11$ | $69.92 \pm 0.17$ | 77.24 |
| FastFISF | $78.94 \pm 1.92$ | $69.42 \pm 1.44$ | $77.14 \pm 0.94$ | $88.10 \pm 1.38$ | $79.09 \pm 1.42$ | $69.53 \pm 0.21$ | 77.04 |

| | | | Uniform missing | | | | |
|---|---|---|---|---|---|---|---|
| Method | CORA | CITESEER | PUBMED | PHOTO | COMPUTERS | OGBN-ARXIV | Average |
| FISF | $79.09 \pm 1.73$ | $69.52 \pm 1.81$ | $77.53 \pm 1.28$ | $88.32 \pm 1.37$ | $82.12 \pm 0.51$ | $69.81 \pm 0.16$ | 77.73 |
| FastFISF | $79.29 \pm 1.84$ | $69.39 \pm 1.57$ | $77.41 \pm 1.77$ | $88.03 \pm 1.46$ | $81.70 \pm 0.54$ | $69.45 \pm 0.18$ | 77.55 |

We can utilize not only channel-wise inter-node diffusion in PCFI but also FP for pre-diffusion. We define FISF using FP for pre-diffusion as FastFISF that can be efficient by using FP without calculation of shortest path distance. Table 9 demonstrates the results of FastFISF compared to the original FISF on semi-supervised node classification tasks. For not low-variance channels, features obtained via pre-diffusion are preserved until diffusion with synthetic features ends. Therefore, since PCFI outperforms FP in terms of performance in downstream tasks, FISF shows slightly better performance than FastFISF in most cases. However, since the performance of FastFISF is comparable to FISF, FastFISF can be a fast variant of FISF without significant performance loss.

## C.6  COMPLEXITY ANALYSIS

Here we discuss the complexity of FISF which involves two diffusion stages: pre-diffusion and diffusion with synthetic features. FISF takes $O(|\mathcal{E}| + (1 + \gamma F)N^2)$ time under structural-missing settings. Under uniform-missing settings, FISF takes $O(|\mathcal{E}| + (1 + \gamma)FN^2)$ time.

We observe that the majority of the computation time in FISF is consumed by employing Dijkstra's algorithm to calculate shortest path distance for each channel. The time complexity of Dijkstra's algorithm is $O(N^2)$. In pre-diffusion under structural missing settings, Dijkstra's algorithm is once utilized since nodes with observed features are equal across all the channels. However, under uniform-missing settings, the time complexity of pre-diffusion increases to $O(N^2F)$, considering the use of Dijkstra's algorithm across all channels.

Table 10: Training time of methods. OOM denotes an out-of-memory error.

| Missing way | structural | | uniform | |
|---|---|---|---|---|
| Method | Cora | PubMed | Cora | PubMed |
| GCNMF | 10.3s | 19.4s | 9.87s | 28.3s |
| GRAFENNE | 47.9s | 74.7s | 51.1s | 74.0s |
| MEGAE | 1753s | OOM | 1801s | OOM |
| FP | 2.36s | 3.12s | 2.25s | 2.90s |
| PCFI | 2.45s | 3.23s | 11.1s | 34.1s |
| FastFISF | 13.4s | 34.6s | 11.8s | 42.5s |
| FISF | 13.4s | 34.8s | 17.6s | 78.2s |

In Appendix C.5, we conduct experiments with FastFISF which is a fast variant of FISF. To address the increasing time complexity in uniform-missing settings, we can employ FastFISF where the time complexity is $O(|\mathcal{E}| + \gamma F N^2)$ regardless of missing way. Therefore, to address the increasing time complexity of FISF in uniform-missing settings, we can employ FastFISF, accompanied by only a slight performance loss.

Table 10 demonstrates the training time of methods. FP has the lowest training time among the methods. However, FISF brings great performance improvement compared to FP. For instance, in structural-missing setups with $r_m = 0.995$, FISF achieves significant gains in node classification accuracy over FP, showing improvements of 7.43% and 4.94% on Cora and PubMed, respectively. We can further confirm that FastFISF significantly decrease the training time in uniform-missing settings.

## C.7 DISCUSSION ON JUSTIFICATION OF SYNTHETIC FEATURE INJECTION

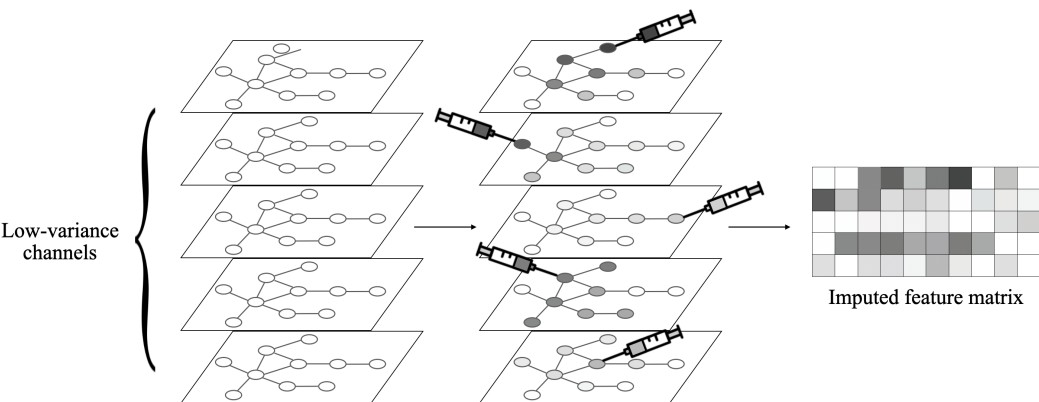

Figure 9: Diffusing a synthetic feature for each low-variance channels results in distinctive imputed features across nodes.

In low-variance channels, all missing features are filled with nearly the same values regardless of connectivity, which can not provide any structural information. In contrast, in our scheme, for each low-variance channel, the synthetic feature diffuses its value to its surroundings and creates a local spike centered on the node with the synthetic features. Each node has larger differences in values from the synthetic feature as the distance from the central node increases. If we inject one synthetic feature into each low variance channel, but place it at a different location for each channel. Then the diffused node feature vector containing every low-variance channel feature after diffusion becomes distinctive from those of the other nodes by reflecting the graph structure. Figure 9 illustrates a visualization of the distinctiveness of the diffused feature vector by our scheme.

# D EXPERIMENTAL DETAILS

## D.1 DATASET DETAILS

Table 11 summarizes the dataset statistics. All the datasets used in this paper are provided in Pytorch Geometric. Following Rossi et al. (2022) and Um et al. (2022), we conduct all experiments on the largest connected graph of each dataset. FISF can also handle disconnected graphs by working on each connected graph.

Table 11: Dataset statistics.

| Dataset | #Nodes | #Edges | #Features | #Classes |
|---|---|---|---|---|
| CORA | 2,485 | 5,069 | 1,433 | 7 |
| CITESEER | 2,120 | 3,679 | 3,703 | 6 |
| PUBMED | 19,717 | 44,324 | 500 | 3 |
| PHOTO | 7,487 | 119,043 | 745 | 8 |
| COMPUTERS | 13,381 | 245,778 | 767 | 10 |
| OGBN-ARXIV | 169,343 | 1,166,243 | 128 | 40 |

## D.2 IMPLEMENTATION DETAILS

We conduct all the experiments on a single NVIDIA GeForce RTX 2080 Ti GPU and an Intel Core I5-6600 CPU @ 3.30 Hz.

**Semi-supervised node classification.** We randomly create 5 different training/validation/test node splits for each dataset except for OGBN-Arxiv. (The node split of OGBN-Arxiv is fixed according to published years of papers (*i.e.,* nodes).) Following the splits in Klicpera et al. (2019), we assign 20 nodes per class as training nodes. Subsequently, the number of validation nodes is adjusted to ensure that when combined with the training nodes, it totals $1,500$. For test nodes, we include all nodes except those designated as training or validation nodes.

Vanilla GCN models for imputation methods (MEGAE (Gao et al., 2023), FP (Rossi et al., 2022), PCFI (Um et al., 2022), and our FISF) and GCNMF models are trained as follows. We utilize Adam optimizer (Kingma & Welling, 2013) and set the maximum number of epochs to $10,000$. We use an early stopping strategy based on validation accuracy, with a patience of 200 epochs. We apply dropout (Srivastava et al., 2014) with the drop probability $p$. $p$ and learning rates in all experiments are searched in $\{0, 0.25, 0.5\}$ and $\{0.01, 0.005, 0.001, 0.0001\}$, respectively, using grid search on validation sets. We train GRAFENNE models by following the training details specified in Gupta et al. (2023).

For all the baselines, we follow all the hyperparameters specified in the original papers or codes. If hyperparameters (specifically, hidden dimension and the number of layers) for a specific dataset are not clarified in the papers, we perform a hyperparameter search using a grid search approach. The search ranges of hidden dimension and the number of layers are $\{16, 32, 64, 128, 256\}$ and $\{2, 3\}$, respectively.

**Link prediction.** We randomly create 5 different training/validation/test edge splits for each dataset. For each split, as the splits in Kipf & Welling (2016b), we assign $10\%$ edges for the training set, $5\%$ edges for the validation set, and $85\%$ edges for the test set.

For GAE models for the imputation methods, we commonly train the models as follows. We use Adam optimizer and set the number of epochs to 200. Learning rates are searched from $\{0.01, 0.005, 0.001, 0.0001\}$ by grid search on validation sets. Following Kipf & Welling (2016b), Taguchi et al. (2021), and Um et al. (2022), we leverage GAE models with 32-dimensional hidden layer and 16-dimensional latent variables.

**FISF implementation.** For semi-supervised node classification tasks, we set the number of layers and learning rates to 64 and 0.005, respectively. For link prediction tasks on Cora, CiteSeer, and PubMed, we set learning rates to 0.01. We set learning rates to 0.001 for Photo and Computers. In all experiments, we fix $K$ to 100 and dropout is applied with $p = 0.5$. In the case of experiments on OGBN-Arxiv, following FP (Rossi et al., 2022) and PCFI (Um et al., 2022), we leverage GCN

Table 12: FISF hyper-parameters used in experiments on semi-supervised node classification tasks.

| Missing way | Structural missing | | | | | | | | | | | | Uniform missing | | |
|---|---|---|---|---|---|---|---|---|---|---|---|---|---|---|---|
| $r_m$ | 0.5 | | | 0.9 | | | 0.995 | | | 0.999 | | | 0.995 | | |
| Datasets | $\alpha$ | $\beta$ | $\gamma$ | $\alpha$ | $\beta$ | $\gamma$ | $\alpha$ | $\beta$ | $\gamma$ | $\alpha$ | $\beta$ | $\gamma$ | $\alpha$ | $\beta$ | $\gamma$ |
| CORA | 0.7 | 0.9 | 50 | 0.9 | 0.7 | 90 | 0.9 | 0.7 | 90 | 0.9 | 0.9 | 90 | 0.9 | 0.9 | 70 |
| CITESEER | 0.7 | 0.7 | 30 | 0.9 | 0.5 | 50 | 0.9 | 0.9 | 90 | 0.9 | 0.9 | 90 | 0.9 | 0.9 | 30 |
| PUBMED | 0.9 | 0.7 | 70 | 0.9 | 0.5 | 10 | 0.9 | 0.5 | 90 | 0.9 | 0.5 | 90 | 0.9 | 0.5 | 90 |
| PHOTO | 0.5 | 0.7 | 90 | 0.1 | 0.9 | 70 | 0.1 | 0.1 | 70 | 0.1 | 0.1 | 50 | 0.1 | 0.1 | 30 |
| COMPUTERS | 0.1 | 0.1 | 90 | 0.1 | 0.7 | 50 | 0.1 | 0.1 | 50 | 0.1 | 0.1 | 90 | 0.1 | 0.5 | 50 |
| OGBN-ARXIV | 0.3 | 0.3 | 10 | 0.1 | 0.3 | 30 | 0.1 | 0.1 | 90 | 0.1 | 0.1 | 70 | 0.1 | 0.1 | 90 |

Table 13: FISF hyper-parameters used in experiments on link prediction tasks.

| Missing way | Structural missing | | | Uniform missing | | |
|---|---|---|---|---|---|---|
| $r_m$ | 0.995 | | | 0.995 | | |
| Datasets | $\alpha$ | $\beta$ | $\gamma$ | $\alpha$ | $\beta$ | $\gamma$ |
| Cora | 0.5 | 0.9 | 90 | 0.3 | 0.9 | 10 |
| CiteSeer | 0.9 | 0.9 | 90 | 0.1 | 0.7 | 10 |
| PubMed | 0.1 | 0.3 | 70 | 0.1 | 0.5 | 90 |
| Computers | 0.1 | 0.9 | 10 | 0.1 | 0.9 | 70 |
| Photo | 0.1 | 0.7 | 10 | 0.1 | 0.7 | 10 |

layers with skip connections (Xu et al., 2018) and set the hidden dimension to 256. Hyperparamters ($\alpha$, $\beta$, and $\gamma$) of FISF used in experiments are summarized in Table 12 and Table 13. We will release the code upon publication.

**Baselines implementation.** For LP, we use codes implemented in Pytorch Geometric (Fey & Lenssen, 2019). The hyperparameter $\alpha$ of LP is searched from $\{0..95, 0.9, 0.8, 0.7, \ldots, 0.1\}$. For the baselines except for LP, we use code released by the authors of papers. The URL links for the baselines are given in Table 14.

Table 14: URL links for baselines.

| Baseline | URL link |
|---|---|
| GCNMF | https://github.com/marblet/GCNmf |
| GRAFENNE | https://github.com/data-iitd/Grafenne |
| MEGAE | https://github.com/zqgao22/max-entropy-gae |
| FP | https://github.com/twitter-research/feature-propagation |
| PCFI | https://github.com/daehoum1/pcfi |

# E ADDITIONAL EXPERIMENTAL RESULTS

## E.1 AP RESULTS ON LINK PREDICTION

Table 15: Performance on link prediction tasks at $r_m = 0.995$, measured in AP (%). Standard deviation errors are given. The best result is highlighted in bold and underlined, while the second-best result is highlighted only in bold. OOM denotes an out-of-memory error.

### Structural missing

| Method | CORA | CITESEER | PUBMED | PHOTO | COMPUTERS |
|---|---|---|---|---|---|
| Full features | $92.62 \pm 1.13$ | $91.60 \pm 1.44$ | $96.59 \pm 0.32$ | $95.24 \pm 0.39$ | $93.77 \pm 0.61$ |
| GCNMF | $70.20 \pm 0.80$ | $69.19 \pm 1.78$ | $86.20 \pm 0.32$ | $80.58 \pm 0.28$ | $83.34 \pm 0.17$ |
| GRAFENNE | $64.70 \pm 3.76$ | $72.08 \pm 9.71$ | $70.43 \pm 3.74$ | $64.78 \pm 0.84$ | $66.56 \pm 1.14$ |
| MEGAE | $69.78 \pm 0.78$ | $70.85 \pm 2.92$ | OOM | $86.46 \pm 1.65$ | $86.12 \pm 1.13$ |
| FP | $86.40 \pm 1.26$ | $82.61 \pm 1.96$ | $83.98 \pm 0.79$ | $93.74 \pm 0.57$ | $91.50 \pm 0.56$ |
| PCFI | $88.63 \pm 0.90$ | $82.98 \pm 0.86$ | $87.07 \pm 0.42$ | $\underline{\mathbf{96.31 \pm 0.25}}$ | $94.58 \pm 0.37$ |
| FISF | $\mathbf{88.81 \pm 1.35}$ | $\underline{\mathbf{85.85 \pm 1.38}}$ | $\mathbf{87.55 \pm 0.35}$ | $95.33 \pm 0.22$ | $\mathbf{94.71 \pm 0.26}$ |
| FISF+NIP | $\underline{\mathbf{89.35 \pm 1.24}}$ | $\mathbf{85.25 \pm 1.85}$ | $\underline{\mathbf{87.62 \pm 0.12}}$ | $\mathbf{95.95 \pm 0.18}$ | $\underline{\mathbf{95.41 \pm 0.33}}$ |

### Uniform missing

| Method | CORA | CITESEER | PUBMED | PHOTO | COMPUTERS |
|---|---|---|---|---|---|
| Full features | $92.62 \pm 1.13$ | $91.60 \pm 1.44$ | $96.59 \pm 0.32$ | $95.24 \pm 0.39$ | $93.77 \pm 0.61$ |
| GCNMF | $64.21 \pm 2.01$ | $65.06 \pm 2.67$ | $82.64 \pm 2.17$ | $80.61 \pm 0.20$ | $83.38 \pm 0.12$ |
| GRAFENNE | $75.04 \pm 13.33$ | $71.39 \pm 9.71$ | $73.56 \pm 5.77$ | $68.36 \pm 7.71$ | $69.79 \pm 5.81$ |
| MEGAE | $67.98 \pm 1.85$ | $63.67 \pm 2.89$ | OOM | $83.22 \pm 1.48$ | $85.11 \pm 2.00$ |
| FP | $88.67 \pm 1.26$ | $85.39 \pm 1.89$ | $82.99 \pm 2.14$ | $95.51 \pm 0.19$ | $94.06 \pm 0.27$ |
| PCFI | $89.13 \pm 1.06$ | $\underline{\mathbf{85.47 \pm 1.82}}$ | $88.20 \pm 0.38$ | $\underline{\mathbf{96.87 \pm 0.20}}$ | $\mathbf{95.55 \pm 0.32}$ |
| FISF | $\mathbf{89.16 \pm 0.77}$ | $\mathbf{85.17 \pm 2.00}$ | $\underline{\mathbf{88.73 \pm 0.36}}$ | $96.27 \pm 0.23$ | $95.12 \pm 0.32$ |
| FISF+NIP | $\underline{\mathbf{89.23 \pm 0.89}}$ | $84.73 \pm 2.00$ | $\mathbf{88.72 \pm 0.36}$ | $\mathbf{96.32 \pm 0.26}$ | $\underline{\mathbf{96.12 \pm 0.30}}$ |

## E.2 DISTRIBUTIONS OF FEATURE VARIANCES ON CORA

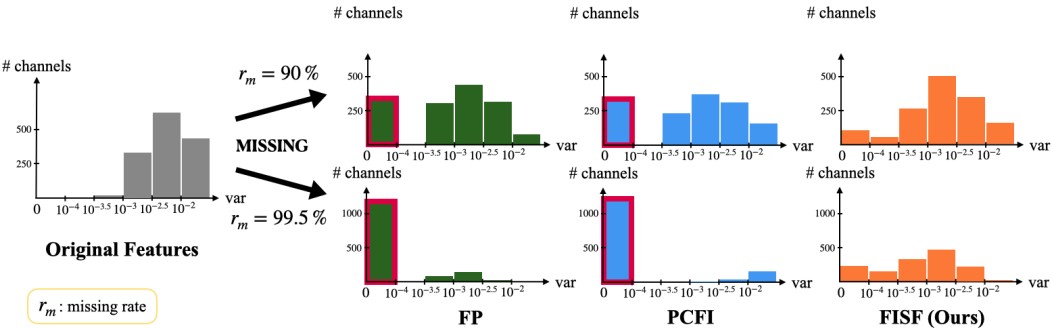

Figure 10: Distributions of variances for each feature channel on Cora dataset with 90%/99.5% missing features. FP and PCFI generates output matrices with many low-variance channels outlined in red, whereas FISF resolves the issue of low-variance channels.

### E.3 ZERO INITIALIZATION VS RANDOM INITIALIZATION

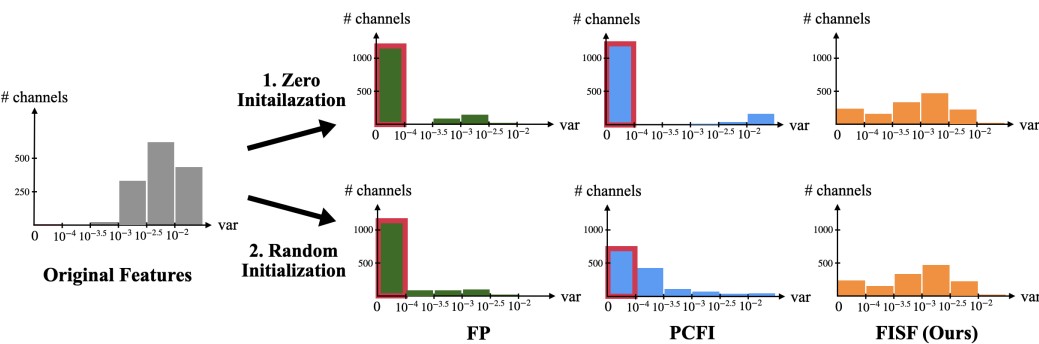

Figure 11: Distributions of variances for each feature channel with zero/random initialization for missing features. Cora dataset with 99.5% missing features is commonly used.

