# OpenReview forum: "Diffusion with Synthetic Features: Feature Imputation for Graphs with Partially Observed Features"
_ICLR.cc/2024/Conference — Submitted to ICLR 2024_

### Official Review · Reviewer_8JVT · 2023-10-28

**Soundness:** 3 good
**Presentation:** 3 good
**Contribution:** 3 good
**Rating:** 6
**Confidence:** 2

**Summary:**

The paper addresses the task of learning on graphs with missing features, specifically focusing on improving the application of graph neural networks (GNNs) to real-world graph-structured data. The paper introduces a novel diffusion-based imputation scheme called Feature Imputation with Synthetic Features (FISF).

(1) It generates synthetic features via pre-diffusion for randomly chosen nodes in these channels.

(2) The diffusion process spreads these synthetic features while also considering observed features simultaneously.

(3) The proposed scheme has been empirically tested, showing promising results, especially in scenarios with a high rate of missing
features. It shows robust performance in both semi-supervised node classification and link prediction tasks.

**Strengths:**

1. The paper addresses a significant and practical problem in the domain of graph learning.

2. The proposed FISF method is novel, focusing on low-variance channels that were overlooked by previous approaches.

3. The paper seems to provide extensive experiments on graphs with varying rates of missing features, demonstrating the robustness of the proposed method.

**Weaknesses:**

1. From the provided content, it's unclear how the proposed method scales with large real-world graphs or its computational efficiency.

2. The abstract and introduction don't mention how generalizable the method is across diverse types of graph-structured data or different

**Questions:**

See weakness

---

> ### Author Response · Authors · 2023-11-21
> **Response to Reviewer 8JVT**
>
> We appreciate your encouragement and hope that the following answers can make you more confident.
>
> > $\textbf{Q1.}$ From the provided content, it's unclear how the proposed method scales with large real-world graphs or its computational efficiency.
>
> FISF operates in structural-missing settings with a time complexity of $O(|\mathcal{E}|+(1+\gamma F)N^2)$ and in uniform-missing settings with a complexity of $O(|\mathcal{E}|+(1+\gamma) F N^2)$. During the rebuttal period, to address the increasing time complexity in uniform-missing settings, we have sought a way to replace channel-wise inter-node diffusion with FP in pre-diffusion as a light version of FISF, called FastFISF. Consequently, the time complexity of FastFISF decreases to $O(|\mathcal{E}|+\gamma FN^2)$ regardless of the missing way.
>
> The table below displays the training time of methods. FP exhibits the shortest training time among the methods. However, FISF notably enhances performance in downstream tasks compared to FP. For instance, in structural-missing setups with $r_m=0.995$, FISF achieves significant gains in node classification accuracy over FP, showing improvements of $7.43$% and $4.94$% on Cora and PubMed, respectively. Additionally, FastFISF demonstrates substantial reductions in training time under uniform-missing settings. A detailed explanation of FastFISF including performance in downstream tasks is in Appendix C.5.
>
> OGBN-Arxiv with ~0.2M nodes is the largest real-world graph dataset in this paper. On OGBN-Arxiv, we show the effectiveness of FISF on semi-supervised node classification tasks and FISF requires only 16.1 seconds for imputation. We agree with the reviewer that the computational efficiency of FISF was unclear. Hence, we added complexity analysis in Appendix C.6.
>
> <Table A: Training time of methods. OOM denotes an out-of-memory error.>
>
> | Missing way | Structural | missing   | Uniform | missing |
> |-------------|:----------:|:---------:|:-------:|:-------:|
> | Method      |    Cora    |   PubMed  |   Cora  |  PubMed |
> | GCNMF       |   $10.3$s  |  $19.4$s  | $9.87$s | $28.3$s |
> | GRAFENNE    |   $47.9$s  |  $74.7$s  | $51.1$s | $74.0$s |
> | MEGAE       |   $1753$s  |    OOM    | $1801$s |   OOM   |
> | FP          |   $2.36$s  |  $3.12$s  | $2.25$s | $2.90$s |
> | PCFI        |   $2.45$s  |  $3.23$s  | $11.1$s | $34.1$s |
> | FastFISF    |   $13.4$s  |  $34.6$s  | $11.8$s | $42.5$s |
> | FISF        |   $13.4$s  |  $34.8$s  | $17.6$s | $78.2$s |
>
>
> > $\textbf{Q2.}$ The abstract and introduction don't mention how generalizable the method is across diverse types of graph-structured data or different.
>
> FISF has good generalizability across diverse types of graph-structured data, extending its applicability to both  hypergraphs and heterogeneous graphs. In case of hypergraphs, a hypergraph can be transformed into a homogeneous graph via clique expansion. Therefore, missing features of nodes in a hypergraph can be also imputed by FISF. In case of heterogeneous graphs, our FISF can be applied to heterogeneous graphs throughout  meta-paths. However, since this application is not straightforward, we mention these extensions as future work in Conclusion.

---

> > ### Author Response · Authors · 2023-11-23
> > **Official Comment by Authors**
> >
> > Dear Reviewer 8JVT,
> >
> > We're grateful for your feedback on our work. As the discussion period nears its end, we would like to confirm if our responses have sufficiently clarified and addressed your concerns. We are happy to provide any additional clarification and discussion.
> >
> > Thank you.

---

### Official Review · Reviewer_VoZJ · 2023-10-30

**Soundness:** 2 fair
**Presentation:** 3 good
**Contribution:** 2 fair
**Rating:** 6
**Confidence:** 4

**Summary:**

This paper claims that existing methods output low variance channels within imputed features and then presents a method called FISF, which performs diffusion with randomly injected synthetic features on low variance channels discovered from pre-diffusion process.

**Strengths:**

- This paper empirically shows that existing diffusion-based methods causes many low variance channels within imputed features, while proposed method FISF can solve those problem (Figure 1).
- The paper is well-written and easy to follow.

**Weaknesses:**

-	This paper does not clearly justify why alleviating the low variance channel problem of the existing diffusion methods is significant for graph learning tasks, even though it is a key motivation of this paper. Without additional evidences, it is hard to agree with this paper’s argument that low variance channels contribute very little to performance.
-	This paper lacks justification of using naïve random synthetic features. Authors claim that more distinctive representations are crucial for classification tasks, but I have concern that randomly injected synthetic features can lead to lower intra-class node representation similarity thus can be harmful on downstream tasks. In my opinion, discussion on aforementioned concern is required.
-	It is required to include the time complexity of presented method. Since V_{k}^{(a)} differs by channel ‘a’ as synthetic feature-injected node differs by channels, diffusion process should be done by channels. Moreover, as presented method further requires pre-diffusion process, presented FISF seems to have much higher time complexity compared to existing methods, especially FP.
-	As presented in the Table 5 in the Appendix, hyperparameters have been tuned with respect to each feature missing rate r_m. Regarding complexity of setting optimal hyperparameters, there is some concern about practical usage of FISF.
-	Lastly, considering the similarity of Label Propagation (LP) and FP, and the argument about low variance channels (in respect of very high missing feature rates), I think this paper shares similar motivation with Poisson Learning [1], which mitigates very low label rate problem in LP. Poisson Learning points out that LP outputs almost constant pseudo-labels for most of unlabeled samples, while this paper points out existing diffusion-based methods including FP causes “low variance channels”, i.e. almost constant features (or an exact constant when there is only one observed feature value, as presented in this paper). Regarding aforementioned similarity, further discussion with Poisson Learning might be beneficial.

**Questions:**

-	Can you provide the feature variance distribution (Figure 1) in more realistic missing feature settings (I.e. r_m <= 0.9)? In my opinion, authors have to show that existing diffusion-based methods still suffer from the problem of making low variance channels in various missing feature settings, especially in more realistic scenarios.
-	Can you provide further theoretical or empirical evidences that can explain why random synthetic feature works well? (to resolve concern in W2)
-	What happens if pre-diffusion method is replaced to FP from PCFI? (Why this paper have chosen PCFI as a pre-diffusion method?)

[1] Jeff Calder, Brendan Cook, Matthew Thorpe and Dejan Slepcev. Poisson Learning: Graph Based Semi-Supervised Learning At Very Low Label Rates. In International Conference on Machine Learning, pp. 1306-1316. PMLR, 2020.

---

> ### Author Response · Authors · 2023-11-21
> **Response to Reviewer VoZJ [Part 1/5]**
>
> Thank you for the time and valuable feedback. We provide the answers below.
>
> > $\textbf{Q1.}$ This paper does not clearly justify why alleviating the low variance channel problem of the existing diffusion methods is significant for graph learning tasks, even though it is a key motivation of this paper. Without additional evidences, it is hard to agree with this paper’s argument that low variance channels contribute very little to performance.
>
> $\textbf{A1.}$ Since low-variance channels do not have any distinctive features, they contribute very little to performance in downstream tasks. To justify addressing the low-variance channel problem, we carry out extra experiments that provide evidence that low-variance channels have little contribution to downstream tasks. We compare performance by removing partial channels from an initial feature matrix using two distinct ways. One method involves excluding channels in descending order of variance, beginning with the highest and based on a fixed percentage. The second way is excluding channels in ascending order of variance. As demonstrated in Figure 5 and Figure 6 in Appendix C.3, performance remains relatively steady even as we remove an increasing proportion of low-variance channels. However, removing channels starting from the highest variance results in notable performance drops, even with a small proportion of channel removal.
>
> > $\textbf{Q2.}$ This paper lacks justification of using naïve random synthetic features. Authors claim that more distinctive representations are crucial for classification tasks, but I have concern that randomly injected synthetic features can lead to lower intra-class node representation similarity thus can be harmful on downstream tasks. In my opinion, discussion on the aforementioned concern is required. Can you provide further theoretical or empirical evidences that can explain why random synthetic feature works well?
>
> $\textbf{A2.}$ To provide evidence regarding smoothness and intra-class node feature similarity, we conduct additional experiments. First, we measure Dirichlet energy of imputed features to compare graph-level smoothness. The table below shows that FISF gives the highest Dirichlet energy (distinctiveness) among the imputation methods.
>
> <Table A: log($E_D$) of imputed features under a structural-missing setting with $r_m=0.995$, where $E_D$ is the Dirichlet energy. Original denotes original given features.>
>
> |     Missing way     |        | Structural |         |        |  Uniform |         |
> |:-------------------:|:------:|:----------:|:-------:|:------:|:--------:|:-------:|
> | Method $\downarrow$ |  Cora  |  CiteSeer  |  PubMed |  Cora  | CiteSeer |  PubMed |
> | Original            | $4.36$ |   $4.49$   |  $3.11$ | $4.36$ |  $4.49$  |  $3.11$ |
> | FP                  | $1.90$ |   $1.94$   | $0.798$ | $1.89$ |  $1.91$  | $0.805$ |
> | PCFI                | $3.14$ |   $2.59$   |  $1.49$ | $2.52$ |  $2.64$  |  $1.43$ |
> | FISF (Ours)         | $3.25$ |   $2.92$   |  $4.15$ | $2.69$ |  $2.70$  |  $4.34$ |
>
> In addition, we conduct further experiments to investigate smoothness within classes. The table below demonstrates the intra-class cosine similarity calculated from imputed features by FISF. Ratio denotes average intra-class similarity/inter-class similarity. If Ratio is greater than 1, inter-class similarity becomes less than the average intra-class similarity, which means the feature is distinctive enough for classification of node features.  We include t-SNE plots that visualize imputed features and deep features in Appendix C.4. We verify that FISF provides clearer cluster structures for both imputed features and deep features than the other imputation methods. Since the imputed features still give distinctiveness and improve the classification performance, we can say that the randomly injected synthetic features are not  harmful on downstream tasks.
>
> <Table B: Average cosine similarity of imputed features by FISF, under a structural-missing setting with $r_m=0.995$.>
>
> | Dataset  | Inter-class | class 1 | class 2 | class 3 | class 4 | class 5 | class 6 | class 7 | Average intra-class |  Ratio |
> |----------|:-----------:|:-------:|:-------:|:-------:|:-------:|:-------:|:-------:|:-------:|:-------------------:|:------:|
> | Cora     |   $0.760$   | $0.858$ | $0.902$ | $0.902$ | $0.844$ | $0.691$ | $0.826$ | $0.870$ |       $0.842$       | $1.11$ |
> | CiteSeer |   $0.279$   | $0.267$ | $0.341$ | $0.636$ | $0.282$ | $0.513$ | $0.380$ |    -    |       $0.403$       | $1.45$ |
> | PubMed   |   $0.871$   | $0.893$ | $0.936$ | $0.880$ |    -    |    -    |    -    |    -    |       $0.903$       | $1.04$ |

---

> ### Author Response · Authors · 2023-11-21
> **Response to Reviewer VoZJ [Part 2/5]**
>
> > $\textbf{Q3.}$ It is required to include the time complexity of presented method. Since V_{k}^{(a)} differs by channel ‘a’ as synthetic feature-injected node differs by channels, diffusion process should be done by channels. Moreover, as presented method further requires pre-diffusion process, presented FISF seems to have much higher time complexity compared to existing methods, especially FP.
>
> $\textbf{A3.}$ FISF operates in structural-missing settings with a time complexity of $O(|\mathcal{E}|+(1+\gamma F)N^2)$ and in uniform-missing settings with a complexity of $O(|\mathcal{E}|+(1+\gamma) F N^2)$. During the rebuttal period, to address the increasing time complexity in uniform-missing settings, we have sought a way to replace channel-wise inter-node diffusion with FP in pre-diffusion as a light version of FISF, called FastFISF. Consequently, the time complexity of FastFISF decreases to $O(|\mathcal{E}|+\gamma FN^2)$ regardless of the missing way.
>
> The table below displays the training time of methods. FP exhibits the shortest training time among the methods. However, FISF notably enhances performance in downstream tasks compared to FP. For instance, in structural-missing setups with $r_m=0.995$, FISF achieves significant gains in node classification accuracy over FP, showing improvements of $7.43$% and $4.94$% on Cora and PubMed, respectively. Additionally, FastFISF demonstrates substantial reductions in training time under uniform-missing settings. A detailed explanation of FastFISF including performance in downstream tasks is in Appendix C.5.
>
> <Table C: Training time of methods. OOM denotes an out-of-memory error.>
>
> | Missing way | Structural | missing   | Uniform | missing |
> |-------------|:----------:|:---------:|:-------:|:-------:|
> | Method      |    Cora    |   PubMed  |   Cora  |  PubMed |
> | GCNMF       |   $10.3$s  |  $19.4$s  | $9.87$s | $28.3$s |
> | GRAFENNE    |   $47.9$s  |  $74.7$s  | $51.1$s | $74.0$s |
> | MEGAE       |   $1753$s  |    OOM    | $1801$s |   OOM   |
> | FP          |   $2.36$s  |  $3.12$s  | $2.25$s | $2.90$s |
> | PCFI        |   $2.45$s  |  $3.23$s  | $11.1$s | $34.1$s |
> | FastFISF    |   $13.4$s  |  $34.6$s  | $11.8$s | $42.5$s |
> | FISF        |   $13.4$s  |  $34.8$s  | $17.6$s | $78.2$s |

---

> ### Author Response · Authors · 2023-11-21
> **Response to Reviewer VoZJ [Part 3/5]**
>
> > $\textbf{Q4.}$ As presented in the Table 5 in the Appendix, hyperparameters have been tuned with respect to each feature missing rate r_m. Regarding complexity of setting optimal hyperparameters, there is some concern about practical usage of FISF.
>
> $\textbf{A4.}$ Despite outperforming performance of FISF, conducting a hyperparameter search for FISF with three hyperparameters ($\alpha$, $\beta$, and $\gamma$) can be burdensome in certain situations. However, both $\alpha$ and $\beta$ ($0<\alpha,\beta<1$) play a similar role in a base of distance during calculating PC (\textit{i.e.} $\xi^*_{i,b}=\alpha^{\mathbf{S}^*_{i,b}}$ and $\xi^s_{i,a}=\beta^{\mathbf{S}^s_{i,a}}$). Thus we can combine them into one, i.e., $\alpha = \beta$. By doing this, the search complexity can be reduced from $5^3$ to $5^2$ without the performance degradation by setting five search points for each hyperparameter. The tables below show that the FISF* with the light search does not degrade performance on semi-supervised node classification and link prediction. The version with the light search requires from 20 minutes to 10 hours depending on the datasets, therefore this burden is manageable for practical usage of FISF.
>
> <Table D: Performance on semi-supervised node classification tasks under structural-missing settings with $r_m=0.995$, measured in accuracy (%)>
>
> | Method |      Cora      | CiteSeer | PubMed | Photo | Computers | OGBN-Arxiv | Average |
> |--------|:--------------:|:--------:|:------:|:-----:|:---------:|:----------:|:-------:|
> | FISF   | $79.29\pm1.72$ |     $69.68\pm2.47$     |    $76.90\pm1.50$    |    $88.22\pm0.79$   |      $79.40\pm1.11$     |       $69.92\pm0.17$     |    $77.24$     |
> | FISF*  |         $78.68\pm1.72$        |     $69.68\pm2.47$     | $76.74\pm1.84$       |   $88.22\pm0.79$   |      $79.40\pm1.11$     |    $69.92\pm0.17$        |    $77.11$     |
>
> <Table E: Performance on semi-supervised node classification tasks under uniform-missing settings with $r_m=0.995$, measured in accuracy (%)>
>
> | Method |      Cora      | CiteSeer | PubMed | Photo | Computers | OGBN-Arxiv | Average |
> |--------|:--------------:|:--------:|:------:|:-----:|:---------:|:----------:|:-------:|
> | FISF   | $79.09\pm1.73$ |     $69.52\pm1.81$      |     $77.53\pm1.28$   |    $88.32\pm1.37$    |     $82.12\pm0.51$                  |     $69.81\pm0.16$    | $77.73$  |
> | FISF*  |       $79.09\pm1.73$         |      $69.52\pm1.81$    |            $76.89\pm2.01$   |      $88.32\pm1.37$     |     $81.56\pm0.47$       |      $69.81\pm0.16$   | $77.53$ |
>
> <Table F: Performance on link prediction tasks under structural-missing settings with $r_m=0.995$, measured in ROC AUC score (%)>
>
> | Method |      Cora      | CiteSeer | PubMed | Photo | Computers| Average |
> |--------|:--------------:|:--------:|:------:|:-----:|:---------:|:----------:|
> | FISF   | $87.26\pm1.44$   | $84.12\pm1.17$       | $83.19\pm0.78$     | $95.86\pm0.21$    | $94.70\pm0.30$  | $89.03$      |
> | FISF*  |         $86.80\pm1.27$   | $84.12\pm1.17$       | $82.46\pm0.94$     | $95.76\pm0.33$    | $94.39\pm0.82$ | $88.70$     |
>
> <Table G: Performance on link prediction tasks under uniform-missing settings with $r_m=0.995$, measured in ROC AUC score (%)>
>
> | Method |      Cora      | CiteSeer | PubMed | Photo | Computers| Average |
> |--------|:--------------:|:--------:|:------:|:-----:|:---------:|:----------:|
> | FISF   | $87.44\pm0.80$   | $83.45\pm2.53$       | $85.33\pm0.47$     | $96.64\pm0.18$    | $95.13\pm0.35$ | $89.60$      |
> | FISF*  |         $87.56\pm1.29$   | $81.15\pm1.17$       | $82.46\pm0.69$     | $95.68\pm0.42$    | $94.94\pm0.27$ | $88.36$     |

---

> ### Author Response · Authors · 2023-11-21
> **Response to Reviewer VoZJ [Part 4/5]**
>
> > $\textbf{Q5.}$ Lastly, considering the similarity of Label Propagation (LP) and FP, and the argument about low variance channels (in respect of very high missing feature rates), I think this paper shares similar motivation with Poisson Learning [1], which mitigates very low label rate problem in LP. Poisson Learning points out that LP outputs almost constant pseudo-labels for most of unlabeled samples, while this paper points out existing diffusion-based methods including FP causes “low variance channels”, i.e. almost constant features (or an exact constant when there is only one observed feature value, as presented in this paper). Regarding aforementioned similarity, further discussion with Poisson Learning might be beneficial.
>
> $\textbf{A5.}$ We thank the reviewer for pointing out a relevant reference [1]. As the reviewer mentioned, our paper and [1] commonly address problems in Laplacian learning adopted by both LP and FP. As you recommended, we added further discussion with Poisson learning [1] to the related work in the revised manuscript as follows. ``The problems being addressed are different, and the causes of each problem are completely different. In [1], proposed Poisson learning is a feature-agnostic method that only propagates given labels like LP, tackling semi-supervised classification. Furthermore, the problem addressed in [1] arises from the very narrow area of a localized spike, generated by the propagation of a given label. The problem assumes having a wide variety of labels evenly distributed despite very low label rates. However, we discover and address the problem of low-variance channels caused by nearly identical observed values with a feature channel. We provide theoretical proof about the cause of the problem of low-variance channels.”
>
> [1] Calder, Jeff, et al. "Poisson learning: Graph based semi-supervised learning at very low label rates." International Conference on Machine Learning. PMLR, 2020.
>
> > $\textbf{Q6.}$ Can you provide the feature variance distribution (Figure 1) in more realistic missing feature settings (I.e. r_m <= 0.9)? In my opinion, authors have to show that existing diffusion-based methods still suffer from the problem of making low variance channels in various missing feature settings, especially in more realistic scenarios.
>
> $\textbf{A6.}$ In the revised version, we provided distributions of feature variances for each channel on CiteSeer containing 90\% missing features in Figure 1. We also added distributions of feature variances on Cora in Appendix E.2. We can observe that the problem of low-variance channels is not limited to high $r_m$ and also occurs in more realistic missing feature settings. We further demonstrate variance distributions of original features without any missing. We confirm that the original features only contain a few low-variance channels like output features from FISF.

---

> ### Author Response · Authors · 2023-11-21
> **Response to Reviewer VoZJ [Part 5/5]**
>
> > $\textbf{Q7.}$ What happens if pre-diffusion method is replaced to FP from PCFI? (Why this paper have chosen PCFI as a pre-diffusion method?)
>
> $\textbf{A7.}$ We select channel-wise inter-node diffusion in PCFI as pre-diffusion to maximize performance in downstream tasks. For not low-variance channels, features obtained via pre-diffusion are preserved until diffusion with synthetic features ends. Therefore, since PCFI outperforms FP in terms of performance in downstream tasks, FISF shows slightly better performance than FastFISF in most cases. We define FISF using FP for pre-diffusion as FastFiSF and the table below shows the comparison results in terms of semi-supervised node classification accuracy. While the original FISF outperforms FastFISF, the performance of FastFISF is comparable to FISF.
>
> <Table H: Performance on semi-supervised node classification tasks under structural-missing settings with $r_m=0.995$, measured in accuracy (%)>
>
> | Method |      Cora      | CiteSeer | PubMed | Photo | Computers | OGBN-Arxiv | Average |
> |--------|:--------------:|:--------:|:------:|:-----:|:---------:|:----------:|:-------:|
> | FISF   | $79.29\pm1.72$ |     $69.68\pm2.47$     |    $76.90\pm1.50$    |    $88.22\pm0.79$   |      $79.40\pm1.11$     |       $69.92\pm0.17$     |    $77.24$     |
> | FastFISF  |         $78.94\pm1.92$   | $69.42\pm1.44$       | $77.14\pm0.94$     | $88.10\pm1.38$    | $79.09\pm1.42$        | $69.53\pm0.21$ | $77.04$     |
>
> <Table I: Performance on semi-supervised node classification tasks under uniform-missing settings with $r_m=0.995$, measured in accuracy (%)>
>
> | Method |      Cora      | CiteSeer | PubMed | Photo | Computers | OGBN-Arxiv | Average |
> |--------|:--------------:|:--------:|:------:|:-----:|:---------:|:----------:|:-------:|
> | FISF   | $79.09\pm1.73$ |     $69.52\pm1.81$      |     $77.53\pm1.28$   |    $88.32\pm1.37$    |     $82.12\pm0.51$                  |     $69.81\pm0.16$    | $77.73$  |
> | FastFISF  |       $79.29\pm1.84$   | $69.39\pm1.57$       | $77.41\pm1.77$     | $88.03\pm1.46$    | $81.70\pm0.54$        | $69.45\pm0.18$  | $77.55$ |
>
>
> The advantage of using FastFISF is a reduction in time complexity. Since channel-wise inter-node diffusion requires more time compared to FP in uniform-missing settings, FastFISF decreases the training time in uniform-missing settings. Table C in $\textbf{A3}$ shows the training time of methods.  While FP has the lowest training time among the methods, FISF brings great performance improvement compared to FP. For instance, on Cora and PubMed, FISF demonstrates improvements in node classification accuracy by $7.43$\%p and $4.94$\%p respectively, compared to FP.

---

> ### Comment · Reviewer_VoZJ · 2023-11-22
>
> Thank you for the authors' earnest response. Overall, most of the doubtful points were addressed. Thus, I have raised my ratings.
> The following are the remaining minor issues in your responses.
> - A2: Can you provide the average (inter/intra-class) cosine similarity of original features like the results presented in Table B (Table 7 in the manuscript) for imputed features? In my opinion, comparing the results shown in Table B with those results might be beneficial and help understand the significance of the results of Table B.
> - A5. I appreciate the authors' responses in general. However, the discussion with Poisson learning added to the manuscript seems insufficient because it is abbreviated compared to the authors' responses. Therefore, we request that the authors' responses be reflected in the manuscript as they are.

---

> ### Author Response · Authors · 2023-11-23
> **Second Response to Reviewer VoZJ**
>
> We appreciate the reviewer's decision. Your insightful feedback has further improved our paper.
>
> > $\textbf{Q8.}$ Can you provide the average (inter/intra-class) cosine similarity of original features like the results presented in Table B (Table 7 in the manuscript) for imputed features? In my opinion, comparing the results shown in Table B with those results might be beneficial and help understand the significance of the results of Table B.
>
> $\textbf{A8.}$ Table J shows the intra-class cosine similarity calculated from original features. The results indicate that original features also have values of Ratio greater than 1 across the datasets. This means that the datasets also originally have higher intra-class feature similarity compared to inter-class feature similarity. Despite the introduction of synthetic features during diffusion, as shown in Table B, we can observe that imputed features by our scheme consistently maintains higher intra-class feature similarity than inter-class feature similarity. We included this discussion and Table J in Appendix C.4 of the revised manuscript.
>
> <Table J: Average cosine similarity of original features.>
>
> | Dataset  | Inter-class | class 1 | class 2 | class 3 | class 4 | class 5 | class 6 | class 7 | Intra-class Average |  Ratio |
> |----------|:-----------:|:-------:|:-------:|:-------:|:-------:|:-------:|:-------:|:-------:|:-------------------:|:------:|
> | Cora     |   $0.0578$           | $0.841$       | $0.113$       | $0.0896$       | $0.683$       | $0.0690$       | $0.0853$       | $0.109$       | $0.0883$           | $1.53$ |
> | CiteSeer |   $0.0470$          | $0.655$      | $0.0601$      | $0.0617$      | $0.0650$      | $0.762$  | $0.0581$  | -   | $0.0644$ | $1.37$ |
> | PubMed   |   $0.0719$         | $0.112$    | $0.937$     | $0.0779$    | -    | -    | -    | -    | $0.0946$         | $1.32$ |
>
> > $\textbf{Q9.}$  The discussion with Poisson learning added to the manuscript seems insufficient because it is abbreviated compared to the authors' responses. Therefore, we request that the authors' responses be reflected in the manuscript as they are.
>
> $\textbf{A9.}$ We thank you for the suggestion. We added our full discussion with Poisson learning to the related work in the revised manuscript.
>
> Please let us know if the remaining issues are addressed. If you have any further concerns, we would like to have an opportunity to address them.

---

### Official Review · Reviewer_VMHb · 2023-10-31

**Soundness:** 1 poor
**Presentation:** 2 fair
**Contribution:** 1 poor
**Rating:** 5
**Confidence:** 3

**Summary:**

The paper proposes a method for missing value imputation on top of pseudo-confidence-based feature imputation (PCFI) by manipulating the features with low variance, whose were left behind by PCFI. The idea is to insert another random feature for each low variance.

**Strengths:**

I think the original idea is fair that the paper try to amplify low variance features.

**Weaknesses:**

The direction of amplifying low variance features can be a good idea and if it is done right, there might be an optimal point that maximize performance on top of its baseline.
However, the paper shows the algorithm, how to manipulate the synthetically generated random features without a clue why it has to be done that way, for what purpose. Overall, the paper really proposes a "new method" without actually showing what is the problem it is solving and how the method help achieving the goal.

**Questions:**

What is the idea of a synthetic random feature and how it help improving performance?

---

> ### Author Response · Authors · 2023-11-21
> **Response to Reviewer VMHb**
>
> Thank you for the time and valuable feedback. We provide the answer below.
>
> > $\textbf{Q1.}$ What is the idea of a synthetic random feature and how it help improving performance? A clue why it has to be done that way, for what purpose.
>
> $\textbf{A1.}$ The synthetic random feature is used as a virtual observed feature instead of a randomly chosen  missing feature. This synthetic feature is introduced to change a low variance channel into a high variance channel to increase distinctiveness. The distinctiveness leads to high performance of the downstream tasks.
>
> In Figure 1 in the revised manuscript, we demonstrate that existing diffusion-based imputation methods generate low-variance channels that contribute very little to distinguishing nodes. A low-variance channel occurs when observed features within the channel have nearly identical values. Furthermore, to provide a clue on how synthetic random features improve performance, we conduct additional experiments on real-world datasets (Appendix C.3). We empirically verify that low-variance channels (not distinctive channels) contribute very little to performance in downstream tasks. To make features in low-variance channels help GNNs perform downstream tasks well, we inject synthetic features with randomly sampled values. When a synthetic feature is provided to a low-variance channel, there exists a feature value different from the observed features within the channel. Hence, treating the synthetic feature as a virtual observed feature can make the channel deviate from low variance. We observe that only a few low-variance channels remain in imputed features by our scheme. Across various missing settings, our scheme achieves state-of-the-art performance in both semi-supervised node classification and link prediction.

---

> > ### Comment · Reviewer_VMHb · 2023-11-22
> >
> > Thank you for your clarification.
> >
> > I think I am clear in the fact that the paper add random features to low variance channel that help to distinguish nodes, which is a way of adding more variance to low variance channels. What I still find missing is the justification of how more variance in any channel help to have better performance. One can add any amount of variance to any channel by random features, which increases distinctiveness, there is no clues why it should help any downstream task.
> > I does not see this is a valid argument to explain performances. I keep my evaluation.

---

> ### Author Response · Authors · 2023-11-22
> **Second Response to Reviewer VMHb [Part 1/2]**
>
> We agree with the reviewer’s opinion that injecting many synthetic features into a low-variance channel disrupts the influence of the observed features due to the generation of many artificial spikes. As shown in Table 3 of our ablation study (Appendix C.1), injecting one synthetic feature per channel yields better performance in downstream tasks compared to injecting two synthetic features. We included this discussion on the reason for improving performance in downstream tasks in Appendix C.7. To explain the reason succinctly, we injected one synthetic feature into each low variance channel, but placed it at a different location for each channel. Then the diffused node feature vector containing every low-variance channel feature after diffusion becomes distinctive from others by reflecting the graph structure. We will illustrate a visualization on the distinctiveness of the diffused feature vector by our scheme in Appendix C.7.

---

> ### Author Response · Authors · 2023-11-22
> **Second Response to Reviewer VMHb [Part 2/2]**
>
> We have added Figure 9 (in Appendix C.7), which visualizes the distinctiveness of a node feature vector obtained by our method, to the revised manuscript.
>
> We appreciate your constructive feedback. Please let us know if our responses address your concerns.

---

### Official Review · Reviewer_mYHT · 2023-11-01

**Soundness:** 3 good
**Presentation:** 3 good
**Contribution:** 3 good
**Rating:** 6
**Confidence:** 5

**Summary:**

This paper tackles the missing feature problem in graphs through the lens of low-variance channels. To address this, FISF first pre-diffuses the known features to unknown features and generates a synthetic feature on a specific low-variance channel. Finally, it diffuses the synthetic feature widely, treating it as a known feature throughout the graph. Performance across various missing rates demonstrates the efficacy of FISF.

**Strengths:**

1. The problem of the low-variance channel is interesting and provides a new perspective on the missing feature issue in the graph community.

2. The use of generating synthetic features seems to be a straightforward remedy for the low-variance channel.

3. The paper is well-written and easy to follow.

**Weaknesses:**

1. Although the authors demonstrated the existence of a low-variance channel after current diffusion-based methods, FP and PCFI, the link explaining how these low-variance channels act as a bottleneck for overall performance is not comprehensively provided. For example, the performance of node classification could be provided after excluding some portions of low-variance channels. Additionally, I am curious whether the original variance of the dataset, without any missing features, shows low variance as depicted in Figure 1. In this context, the authors should explain why a low-variance channel is especially burdensome in scenarios with missing features.

2. I wonder if the low-variance problem is due to zero-initialization. In cases of severe missing data and zero-initialization is equipped, the majority of the feature matrix would consist of zeros, so the output matrix would naturally contain many zeros (i.e., biases, if adding biases is enabled), especially considering that most graph datasets use one-hot encoding via bag-of-words for feature matrices. If the initialization for the missing feature were from random sampling, such as a uniform or normal distribution, the low variance problem might be easily addressed.

3. Although diffusion via synthetic features can enhance the distinctiveness across features, it might undermine the GNN's key inductive bias, which is the smoothness across connected nodes. A more in-depth discussion of the trade-off between feature distinctiveness and the smoothness of connected features should be provided.

4. While the authors proposed the use of synthetic features, excluding this module, the pre-diffusion and diffusion with synthetic features are exactly aligned with the existing work, PCFI. This raises concerns about the overall novelty of this paper.

5. The complexity of FISF compared to existing works is not comprehensively addressed. Given that the adjacency matrix is created for each feature dimension, the complexity would be exceedingly high, potentially limiting the practical application of FISF.

6. The proposed missing rates of 0.995 and 0.999 seem unrealistic. Furthermore, edge information can also be missing in real-world scenarios, a factor that should be considered.

**Questions:**

See the Weaknesses.

---

> ### Author Response · Authors · 2023-11-21
> **Response to Reviewer mYHT [Part 1/3]**
>
> Thank you for the time and valuable feedback. We provide the answers below.
>
> > $\textbf{Q1.}$ The link explaining how these low-variance channels act as a bottleneck for overall performance is not comprehensively provided. Additionally, I am curious whether the original variance of the dataset, without any missing features, shows low variance as depicted in Figure 1.
>
> $\textbf{A1.}$ We conduct additional experiments to demonstrate little contribution of low-variance channels in downstream tasks. We compare performance by excluding partial channels from an original feature matrix using two different ways. The first way is excluding channels in descending order of variance, starting from the highest, based on a fixed proportion. The second way is excluding channels in ascending order of variance. As shown in Figure 5 and Figure 6 in Appendix C.3, the performance persists despite an increasing removal proportion of low-variance channels. However, cases of channel removal from the highest variance suffer significant performance degradation even with a low proportion of channel removal.
>
> We added the original variance of the datasets before missing, to Figure 1 in the revised manuscript and Figure 9 in Appendix E. Through the original variance of the dataset, we can confirm that the original dataset also contains a few low-variance channels like output features from FISF.
>
> > $\textbf{Q2.}$ I wonder if the low-variance problem is due to zero-initialization. In cases of severe missing data and zero-initialization is equipped, the majority of the feature matrix would consist of zeros, so the output matrix would naturally contain many zeros (i.e., biases, if adding biases is enabled), especially considering that most graph datasets use one-hot encoding via bag-of-words for feature matrices.  If the initialization for the missing feature were from random sampling, such as a uniform or normal distribution, the low variance problem might be easily addressed.
>
> $\textbf{A2.}$ The low-variance channels in an output matrix are not caused by zero initialization for unknown features. According to the proof in Appendix A, initial values for unknown features do not affect the steady state of diffusion. During the diffusion, observed features are preserved by resetting to their original values at every iteration while missing features are updated by aggregating features from neighboring nodes. Eventually, the missing features converge to a steady state depending on only the observed features regardless of the initial values of missing features. That is, the influence of the initial values of the missing features converge to zero.
>
> In the case of Cora, CiteSeer, and PubMed, which are sparse datasets containing many zeros in their original feature matrices, many low (almost zero)-variance channels can occur due to nearly identical observed values within a channel, resulting in a low-variance channel. This phenomenon due to sparsity is entirely different from the zero initialization of missing features. In the case of Photo, Computers, and OGBN-Arxiv, the ratio of zeros within the feature matrix is not high. For example, in OGBN-Arxiv, only 0.0002% of features are zeros since features are obtained by 128-dimensional deep embeddings of words. For all the aforementioned datasets, FISF improves the performance in downstream tasks by addressing low-variance channels regardless of their frequent values.

---

> ### Author Response · Authors · 2023-11-21
> **Response to Reviewer mYHT [Part 2/3]**
>
> > $\textbf{Q3.}$ A more in-depth discussion of the trade-off between feature distinctiveness and the smoothness of connected features should be provided.
>
> $\textbf{A3.}$ As the reviewer mentioned, there exists a trade-off between feature distinctiveness and smoothness on a graph. We discussed feature distinctiveness and smoothness in depth through various experiments. Initially, to explore the graph-level smoothness, we measure Dirichlet energy on imputed features and original features. As shown in the table below, FP designed to minimize Dirichlet energy via diffusion demonstrates the lowest Dirichlet energy. In contrast, FISF makes Dirichlet energy of the imputed features similar to that of the original features without considering the trade-off. Note that our FISF shows the highest Dirichlet energy (distinctiveness) among the methods.
>
> <Table A: log($E_D$) of imputed features under a structural-missing setting with $r_m=0.995$, where $E_D$ is the Dirichlet energy. Original denotes original given features.>
>
> |     Missing way     |        | Structural |         |        |  Uniform |         |
> |:-------------------:|:------:|:----------:|:-------:|:------:|:--------:|:-------:|
> | Method $\downarrow$ |  Cora  |  CiteSeer  |  PubMed |  Cora  | CiteSeer |  PubMed |
> | Original            | $4.36$ |   $4.49$   |  $3.11$ | $4.36$ |  $4.49$  |  $3.11$ |
> | FP                  | $1.90$ |   $1.94$   | $0.798$ | $1.89$ |  $1.91$  | $0.805$ |
> | PCFI                | $3.14$ |   $2.59$   |  $1.49$ | $2.52$ |  $2.64$  |  $1.43$ |
> | FISF (Ours)         | $3.25$ |   $2.92$   |  $4.15$ | $2.69$ |  $2.70$  |  $4.34$ |
>
> We conduct further qualitative analysis on imputed and deep features to examine feature distinctiveness. Figures 7 and 8 in Appendix C.4 depict t-SNE plots visualizing imputed and deep features. These plots indicate that FISF provides clearer cluster structures for both imputed and deep features compared to other imputation methods. While smoothness is an important inductive bias for GNNs, our experimental results confirm that features displaying excessive smoothness lead to poor performance in downstream tasks.
>
> > $\textbf{Q4.}$  The pre-diffusion and diffusion with synthetic features are exactly aligned with the existing work, PCFI.
>
> $\textbf{A4.}$ As the pre-diffusion stage in FISF, we adopt channel-wise inter-node diffusion in PCFI. However, the second diffusion stage involving synthetic features is a new scheme, distinct from PCFI.  Beyond addressing the problem of low-variance channels by generating synthetic features, we introduce a new diffusion stage that employs two distinct types of distance encoding. By combining these two distances, synthetic features exert a stronger influence than observed features during the diffusion process. As a result, FISF outperforms PCFI by a large margin at high rates of missing features.
>
> > $\textbf{Q5.}$  The complexity of FISF compared to existing works.
>
> $\textbf{A5.}$ FISF operates in structural-missing settings with a time complexity of $O(|\mathcal{E}|+(1+\gamma F)N^2)$ and in uniform-missing settings with a complexity of $O(|\mathcal{E}|+(1+\gamma) F N^2)$. During the rebuttal period, to address the increasing time complexity in uniform-missing settings, we have sought a way to replace channel-wise inter-node diffusion with FP in pre-diffusion as a light version of FISF, called FastFISF. Consequently, the time complexity of FastFISF decreases to $O(|\mathcal{E}|+\gamma FN^2)$ regardless of the missing way.
>
> The table below displays the training time of methods. FP exhibits the shortest training time among the methods. However, FISF notably enhances performance in downstream tasks compared to FP. For instance, in structural-missing setups with $r_m=0.995$, FISF achieves significant gains in node classification accuracy over FP, showing improvements of $7.43$% and $4.94$% on Cora and PubMed, respectively. Additionally, FastFISF demonstrates substantial reductions in training time under uniform-missing settings. A detailed explanation of FastFISF including performance in downstream tasks is in Appendix C.5.
>
> <Table B: Training time of methods. OOM denotes an out-of-memory error.>
>
> | Missing way | Structural | missing | Uniform | missing |
> |-------------|:----------:|:---------:|:-------:|:-------:|
> | Method      |    Cora    |   PubMed  |   Cora  |  PubMed |
> | GCNMF       |   $10.3$s  |  $19.4$s  | $9.87$s | $28.3$s |
> | GRAFENNE    |   $47.9$s  |  $74.7$s  | $51.1$s | $74.0$s |
> | MEGAE       |   $1753$s  |    OOM    | $1801$s |   OOM   |
> | FP          |   $2.36$s  |  $3.12$s  | $2.25$s | $2.90$s |
> | PCFI        |   $2.45$s  |  $3.23$s  | $11.1$s | $34.1$s |
> | FastFISF    |   $13.4$s  |  $34.6$s  | $11.8$s | $42.5$s |
> | FISF        |   $13.4$s  |  $34.8$s  | $17.6$s | $78.2$s |

---

> ### Author Response · Authors · 2023-11-21
> **Response to Reviewer mYHT [Part 3/3]**
>
> > $\textbf{Q6.}$ The proposed missing rates of 0.995 and 0.999 seem unrealistic. Furthermore, edge information can also be missing in real-world scenarios, a factor that should be considered.
>
> $\textbf{A6.}$ Research aimed at handling extremely high rates of missing data is actively progressing across various fields (e.g., 99.99% missing in electronic health records [1], 97.5% missing in semiconductor manufacturing data, and 95% missing in an image [3]). In line with this trend, diffusion-based imputation methods have been addressing challenging missing scenarios with a missing rate of 99%/99.5% [4, 5]. To demonstrate that our scheme endures even in more extreme missing scenarios, we include experimental results under 99.9% missing settings.
>
> As an extreme scenario, during the early stages when few people (e.g., 0.1%) purchase a new product, most people (e.g., 99.9%) linked in a social network, having similar purchase tendencies, do not have any features related to this product. In this case, from product-specific features of the very few early adaptors, the proposed diffusion scheme can impute the features. Then, the fully filled matrix can be used as an input by GNNs for a learning task of product recommendation.
>
> We concur with the reviewer’s point that connectivity between entities is not provided in many real-world scenarios. To use graph-based methods in such scenarios, several efforts have been made to form graphs by using the k-NN algorithm [6, 7].
>
> [1] Kim, Yeo-Jin, and Min Chi. "Temporal Belief Memory: Imputing Missing Data during RNN Training." In Proceedings of the 27th International Joint Conference on Artificial Intelligence (IJCAI-2018). 2018\
> [2] Park, Sewon, et al. "Bayesian nonparametric classification for incomplete data with a high missing rate: an application to semiconductor manufacturing data." IEEE Transactions on Semiconductor Manufacturing (2023).\
> [3] Yoon, Seongwook, and Sanghoon Sull. "GAMIN: Generative adversarial multiple imputation network for highly missing data." Proceedings of the IEEE/CVF conference on computer vision and pattern recognition. 2020.\
> [4] Rossi, Emanuele, et al. "On the unreasonable effectiveness of feature propagation in learning on graphs with missing node features." Learning on Graphs Conference. PMLR, 2022.\
> [5] Um, Daeho, et al. "Confidence-Based Feature Imputation for Graphs with Partially Known Features." The Eleventh International Conference on Learning Representations. 2022.\
> [6] Telyatnikov, Lev, and Simone Scardapane. "EGG-GAE: scalable graph neural networks for tabular data imputation." International Conference on Artificial Intelligence and Statistics. PMLR, 2023.\
> [7] Yun, Sukwon, Junseok Lee, and Chanyoung Park. "Single-cell RNA-seq data imputation using Feature Propagation." ICML workshop. 2023.

---

> ### Author Response · Authors · 2023-11-23
> **Official Comment by Authors**
>
> Dear Reviewer mYHT,
>
> We're grateful for your feedback on our work. As the discussion period nears its end, we would like to confirm if our responses have sufficiently clarified and addressed your concerns. We are happy to provide any additional clarification and discussion.
>
> Thank you.

---

> > ### Comment · Reviewer_mYHT · 2023-11-23
> > **Response from Reviewer mYHT**
> >
> > I appreciate the author's detailed rebuttal. The majority of my concerns have been addressed, and I am particularly pleased with the addition of Figure C.3 and the updates made to Figure 1. However, I still have a concern regarding the impact of high missing rates on your model. Specifically, in cases like 99% and 99.9% missing, some columns might consist entirely of zeros, and thus the propagated values would remain zeros.
> >
> > In this regard, could you provide results similar to those in Figure 1 but with random initialization? This would help address my concern. If my concern is satisfactorily resolved, I will directly raise my current score. However, considering the limited remaining time for the rebuttal process, I am prepared to raise my score before the deadline.

---

> > > ### Author Response · Authors · 2023-11-23
> > >
> > > We fixed an issue in Figure 11 and uploaded the revised manuscript.

---

> > > > ### Comment · Reviewer_mYHT · 2023-11-23
> > > >
> > > > Thank you for the last-minute discussion and all the authors' efforts.
> > > >
> > > > I have raised my score to 6.

---

> ### Author Response · Authors · 2023-11-23
> **Second Response to Reviewer mYHT**
>
> Thank you for the positive comments and constructive advice.
>
> To address your remaing concern, we have included Figure 11 in Appendix E.3 of the revised manuscript. Figure 11 compares variance distributions when zero initialization and random initialization are used. Figure 11 shows that many low-varince channels persist despite random initialization, but there is a slight difference between the distributions despite using the same setting. This is because all the diffusion-based methods approximate the steady state with a sufficiently large hyperparameter $K$, indicating the number of diffusion iteration (e.g. $K=40$ is used in FP and $K=100$ is used in PCFI and FISF). However, we have further confirmed that variance distributions becomes identical with very large $K$ values (e.g., $K=1000$) regardless of initialization. Although the final approximated results are not affected by initialization for missing features with a large $K$, careful consideration is needed when determining $K$, depending on the initialization.

---

### Author Response · Authors · 2023-11-21
**General Response to Reviewers**

Dear Reviewers,

We appreciate all reviewers for their constructive feedback and hope that our response convinces the reviewers. Please inform us if the raised issues have been addressed. If the reviewers have additional concerns, we would appreciate the opportunity to further address them. Since we have addressed each review individually, we summarize the most important changes as follows:

1. We included original feature variances of the datasets and distributions of feature variances when the datasets contain $90$% missing features (in Figure 1 and Figure 9 in the revised manuscript).
2. We experimentally confirmed little contribution of low-variance channels in downstream tasks (in Appendix C.3 of the revised manuscript).
3. We analyzed the time complexity of our scheme (in Appendix C.6 of the revised manuscript).

We uploaded the revised manuscript, marking the changes in blue.

---

### Author Response · Authors · 2023-11-23
**Gentle Reminder**

Dear all reviewers,

In response to your reviews, we have conducted additional experiments and revised our manuscript based on the reviews. We have also responded in detail to all of the raised questions.

With the discussion period concluding in 8 hours, we would like to ensure that our responses have adequately clarified and addressed the reviewers' concerns. We are open to providing further clarification and engaging in additional discussions if needed.

Thank you!

---

### Meta-Review · Area_Chair_LFds · 2023-12-15

**Metareview:**

The paper introduces a new method called Feature Imputation with Synthetic Features (FISF) to improve graph neural networks' performance on graphs with missing features. FISF addresses the issue of low-variance channels in previous diffusion-based methods by generating and diffusing synthetic features alongside observed ones. This approach demonstrates enhanced performance in semi-supervised node classification and link prediction tasks, especially in high-missing feature scenarios. Despite its promising results, the paper fails to thoroughly explain the impact of low-variance channels on overall performance, the rationale behind synthetic features, and its computational efficiency, leaving questions about its scalability and applicability to various graph-structured data types.

**Justification For Why Not Higher Score:**

The drawbacks in thoroughly explain the impact of low-variance channels on overall performance.

**Justification For Why Not Lower Score:**

NA

---

### Decision · Program_Chairs · 2024-01-16

Reject